# TEMPERATURE CHECK: THEORY AND PRACTICE FOR TRAINING MODELS WITH SOFTMAX-CROSS-ENTROPY LOSSES

## ABSTRACT

The softmax function combined with a cross-entropy loss is a principled approach to modeling probability distributions that has become ubiquitous in deep learning. The softmax function is defined by a lone hyperparameter, the temperature, that is commonly set to one or regarded as a way to tune model confidence after training; however, less is known about how the temperature impacts training dynamics or generalization performance. In this work we develop a theory of early learning for models trained with softmax-cross-entropy loss and show that the learning dynamics depend crucially on the inverse-temperature $\beta$ as well as the magnitude of the logits at initialization, $||\beta\mathbf{z}||_2$. We follow up these analytic results with a large-scale empirical study of a variety of model architectures trained on CIFAR10, ImageNet, and IMDB sentiment analysis. We find that generalization performance depends strongly on the temperature, but only weakly on the initial logit magnitude. We provide evidence that the dependence of generalization on $\beta$ is not due to changes in model confidence, but is a dynamical phenomenon. It follows that the addition of $\beta$ as a tunable hyperparameter is key to maximizing model performance. Although we find the optimal $\beta$ to be sensitive to the architecture, our results suggest that tuning $\beta$ over the range $10^{-2}$ to $10^1$ improves performance over all architectures studied. We find that smaller $\beta$ may lead to better peak performance at the cost of learning stability.

## 1 INTRODUCTION

Deep learning has led to breakthroughs across a slew of classification tasks (LeCun et al., 1989; Krizhevsky et al., 2012; Zagoruyko and Komodakis, 2017). Crucial components of this success have been the use of the softmax function to model predicted class-probabilities combined with the cross-entropy loss function as a measure of distance between the predicted distribution and the label (Kline and Berardi, 2005; Golik et al., 2013). Significant work has gone into improving the generalization performance of softmax-cross-entropy learning. A particularly successful approach has been to improve overfitting by reducing model confidence; this has been done by regularizing outputs using confidence regularization (Pereyra et al., 2017) or by augmenting data using label smoothing (Müller et al., 2019; Szegedy et al., 2016). Another way to manipulate model confidence is to tune the temperature of the softmax function, which is otherwise commonly set to one. Adjusting the softmax temperature during training has been shown to be important in metric learning (Wu et al., 2018; Zhai and Wu, 2019) and when performing distillation (Hinton et al., 2015); as well as for post-training calibration of prediction probabilities (Platt, 2000; Guo et al., 2017).

The interplay between temperature, learning, and generalization is complex and not well-understood in the general case. Although significant recent theoretical progress has been made understanding generalization and learning in wide neural networks approximated as linear models, analysis of linearized learning dynamics has largely focused on the case of squared error losses (Jacot et al., 2018; Du et al., 2019; Lee et al., 2019; Novak et al., 2019a; Xiao et al., 2019). Infinitely-wide networks trained with softmax-cross-entropy loss have been shown to converge to max-margin classifiers in a particular function space norm (Chizat and Bach, 2020), but timescales of convergence are not known. Additionally, many well-performing models operate best away from the linearized regime (Novak et al., 2019a; Aitchison, 2019). This means that understanding the deviations of

models from their linearization around initialization is important for understanding generalization (Lee et al., 2019; Chizat et al., 2019).

In this paper, we investigate the training of neural networks with softmax-cross-entropy losses. In general this problem is analytically intractable; to make progress we pursue a strategy that combines analytic insights at short times with a comprehensive set of experiments that capture the entirety of training. At short times, models can be understood in terms of a linearization about their initial parameters along with nonlinear corrections. In the linear regime we find that networks trained with different inverse-temperatures, $\beta = 1/T$, behave identically provided the learning rate is scaled as $\tilde{\eta} = \eta\beta^2$. Here, networks begin to learn over a timescale $\tau_z \sim \|\mathbf{Z}^0\|_2/\tilde{\eta}$ where $\mathbf{Z}^0$ are the initial logits of the network after being multiplied by $\beta$. This implies that we expect learning to begin faster for networks with smaller logits. The learning dynamics begin to become nonlinear over another, independent, timescale $\tau_{nl} \sim \beta/\tilde{\eta}$, suggesting more nonlinear learning for small $\beta$. From previous results we expect that neural networks will perform best in this regime where they quickly exit the linear regime (Chizat et al., 2019; Lee et al., 2020; Lewkowycz et al., 2020).

We combine these analytic results with extensive experiments on competitive neural networks across a range of architectures and domains including: Wide Residual networks (Zagoruyko and Komodakis, 2017) on CIFAR10 (Krizhevsky, 2009), ResNet-50 (He et al., 2016) on ImageNet (Deng et al., 2009), and GRUs (Chung et al., 2014) on the IMDB sentiment analysis task (Maas et al., 2011). In the case of residual networks, we consider architectures with and without batch normalization, which can appreciably change the learning dynamics (Ioffe and Szegedy, 2015). For all models studied, we find that generalization performance is poor at $\|\mathbf{Z}^0\|_2 \gg 1$ but otherwise largely independent of $\|\mathbf{Z}^0\|_2$. Moreover, learning becomes slower and less stable at very small $\beta$; indeed, the optimal learning rate scales like $\eta^* \sim 1/\beta$ and the resulting early learning timescale can be written as $\tau_z^* \sim \|\mathbf{Z}^0\|_2/\beta$. For all models studied, we observe strong performance for $\beta \in [10^{-2}, 10^1]$ although the specific optimal $\beta$ is architecture dependent. Emphatically, the optimal $\beta$ is often far from 1. For models without batch normalization, smaller $\beta$ can give stronger results on some training runs, with others failing to train due to instability. Overall, these results suggest that model performance can often be improved by tuning $\beta$ over the range of $[10^{-2}, 10^1]$.

## 2 THEORY

We begin with a precise description of the problem setting before discussing a theory of learning at short times. We will show the following:

- The inverse temperature $\beta$ and logit scale $\|\mathbf{Z}^0\|$ control timescales which determine the rate of change of the loss, the relative change in logits, and the time for learning to leave the linear learning regime.

- Small $\beta$ causes training to access the non-linear learning regime. We will see empirically that increasing access to the non-linear regime can improve generalization.

- The largest allowable learning rate is set by the timescale to leave the linearized learning regime, which suggests that networks with small $\beta$ will train more slowly.

All numerical results in this section are using a Wide Resnet (Zagoruyko and Komodakis, 2017) trained on CIFAR10.

### 2.1 BASIC MODEL AND NOTATION

We consider a classification task with $K$ classes. For an $N$ dimensional input $\mathbf{x}$, let $\mathbf{z}(\mathbf{x}, \boldsymbol{\theta})$ be the pre-softmax output of a classification model parameterized by $\boldsymbol{\theta} \in \mathbb{R}^P$, such that the classifier predicts the class $i$ corresponding to the largest output value $\mathbf{z}_i$. We will mainly consider $\boldsymbol{\theta}$ trained by SGD on a training set $(\mathcal{X}, \mathcal{Y})$ of $M$ input-label pairs. We focus on models trained with cross-entropy loss with a non-trivial *inverse temperature* $\beta$. The softmax-cross-entropy loss can be written as

$$\mathcal{L}(\boldsymbol{\theta}, \mathcal{X}, \mathcal{Y}) = \sum_{i=1}^{K} \mathcal{Y}_i \cdot \ln(\sigma(\beta\mathbf{z}_i(\mathcal{X}, \boldsymbol{\theta}))) = \sum_{i=1}^{K} \mathcal{Y}_i \cdot \ln(\sigma(\mathbf{Z}_i(\mathcal{X}, \boldsymbol{\theta}))) \tag{1}$$

where we define the *rescaled logits* $\mathbf{Z} = \beta\mathbf{z}$ and $\sigma(\mathbf{Z})_i = e^{\mathbf{Z}_i}/\sum_j e^{\mathbf{Z}_j}$ is the softmax function. Here $\mathbf{Z}(\mathcal{X}, \boldsymbol{\theta})$ is the $M \times K$ dimensional matrix of rescaled logits on the training set.

As we will see later, the statistics of individual $\sigma(\mathbf{Z})_i$ will have a strong influence on the learning dynamics. While the statistics of $\sigma(\mathbf{Z})_i$ are intractable for intermediate magnitudes, $\|\mathbf{Z}\|_2$, they can be understood in the limits of large and small $\|\mathbf{Z}\|_2$. For a fixed model $\mathbf{z}(\mathbf{x}, \boldsymbol{\theta})$, $\beta$ controls the certainty of the predicted probabilities. Values of $\beta$ such that $\beta \ll 1/\|\mathbf{z}\|_2$ will give small values of $\|\mathbf{Z}\|_2 \ll 1$, and the outputs of the softmax will be close to $1/K$ independent of $i$ (the maximum-entropy distribution on $K$ classes). Larger values of $\beta$ such that $\beta \gg 1/\|\mathbf{z}\|_2$ will lead to large values of $\|\mathbf{Z}\|_2 \gg 1$; the resulting distribution has probability close to 1 on one label, and (exponentially) close to 0 on the others.

The continuous time learning dynamics (exact in the limit of small learning rate) are given by:

$$\dot{\boldsymbol{\theta}} = \eta\beta \sum_{i=1}^{K} \left( \frac{\partial \mathbf{z}_i(\mathcal{X}, \boldsymbol{\theta}(t))}{\partial \boldsymbol{\theta}} \right)^{\mathrm{T}} (\mathcal{Y}_i - \sigma(\mathbf{Z}_i(\mathcal{X}, \boldsymbol{\theta}(t)))) \tag{2}$$

for learning rate $\eta$. We will drop the explicit dependence of $\mathbf{Z}_i$ on $\boldsymbol{\theta}$ from here onward, and we will denote time dependence as $\mathbf{Z}_i(\mathcal{X}, t)$ explicitly where needed.

In function space, the dynamics of the model outputs on an input $\mathbf{x}$ are given by

$$\frac{d\mathbf{z}_i(\mathbf{x})}{dt} = \eta\beta \sum_{j=1}^{K} (\hat{\Theta}_{\boldsymbol{\theta}})_{ij}(\mathbf{x}, \mathcal{X})(\mathcal{Y}_j - \sigma(\mathbf{Z}_j(\mathcal{X}))) \tag{3}$$

where we define the $M \times M \times K \times K$ dimensional tensor $\hat{\Theta}_{\boldsymbol{\theta}}$, the empirical *neural tangent kernel* (NTK), as

$$(\hat{\Theta}_{\boldsymbol{\theta}})_{ij}(\mathbf{x}, \mathcal{X}') \equiv \frac{\partial \mathbf{z}_i(\mathbf{x})}{\partial \boldsymbol{\theta}} \left( \frac{\partial \mathbf{z}_j(\mathcal{X})}{\partial \boldsymbol{\theta}} \right)^{\mathrm{T}} \tag{4}$$

for class indices $i$ and $j$, which is block-diagonal in the infinite-width limit.

From Equation 3, we see that the early-time dynamics in function space depend on $\beta$, the initial softmax input $\mathbf{Z}(\mathcal{X}, 0)$ on the training set, and the initial $\hat{\Theta}_{\boldsymbol{\theta}}$. Changing these observables across a model family will lead to different learning trajectories early in learning. Since significant work has already studied the effects of the NTK, here we focus on the effects of changing $\beta$ and $\|\mathbf{Z}^0\|_F \equiv \|\mathbf{Z}(\mathcal{X}, 0)\|_F$ (the norm of the $M \times K$ dimensional matrix of training logits), independent of $\hat{\Theta}_{\boldsymbol{\theta}}$.

## 2.2 LINEARIZED DYNAMICS

For small changes in $\boldsymbol{\theta}$, the tangent kernel is approximately constant throughout learning (Jacot et al., 2018), and we drop the explicit $\boldsymbol{\theta}$ dependence in this subsection. The linearized dynamics of $\mathbf{z}(\mathbf{x}, t)$ only depend on the initial value of $\hat{\Theta}$ and the $\beta$-scaled logit values $\mathbf{Z}(\mathcal{X}, t)$. This suggests that there is a universal timescale across $\beta$ and $\eta$ which can be used to compare linearized trajectories with different parameter values. Indeed, if we define an *effective learning rate* $\tilde{\eta} \equiv \eta\beta^2$, we have

$$\frac{d\mathbf{Z}_i(\mathbf{x})}{dt} = \tilde{\eta} \sum_{j=1}^{K} (\hat{\Theta})_{ij}(\mathbf{x}, \mathcal{X})(\mathcal{Y}_j - \sigma(\mathbf{Z}_j(\mathcal{X}))) \tag{5}$$

which removes explicit $\beta$ dependence of the dynamics. We note that a similar rescaling exists for the continuous time versions of other optimizers like momentum (Appendix B).

The effective learning rate $\tilde{\eta}$ is useful for understanding the nonlinear dynamics, as plotting learning curves versus $\tilde{\eta}t$ causes early-time collapse for fixed $\mathbf{Z}^0$ across $\beta$ and $\eta$ (Figure 1). We see that there is a strong, monotonic, dependence of the time at which the nonlinear model begins to deviate from its linearization on $\beta$. We will return to and explain this phenomenon in Section 2.4.

Unless otherwise noted, we will analyze all timescales in units of $\tilde{\eta}$ instead of $\eta$, as it will allow for the appropriate early-time comparisons between models with different $\beta$.

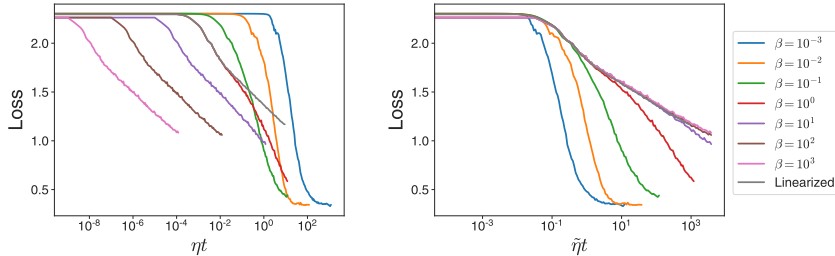

Figure 1: For fixed initial training set logits $\mathbf{Z}^0$, plotting learning curves against $\tilde{\eta}t = \beta^2\eta t$ causes the learning curves to collapse to the learning curve of the linearized model at early times (right), in contrast to un-scaled curves (left). Models with large $\beta$ follow linearized dynamics the longest.

## 2.3 EARLY LEARNING TIMESCALE

We now define and compute the early learning timescale, $\tau_z$, that measures the time it takes for the logits to change significantly from their initial value. Specifically, we define $\tau_z$ such that for $t \ll \tau_z$ we expect $\|\mathbf{Z}(\mathbf{x},t) - \mathbf{Z}(\mathbf{x},0)\|_F \ll \|\mathbf{Z}(\mathbf{x},0)\|_F$ and for $t \gg \tau_z$, $\|\mathbf{Z}(\mathbf{x},t) - \mathbf{Z}(\mathbf{x},0)\|_F \sim \|\mathbf{Z}(\mathbf{x},0)\|_F$ (or larger). This is synonymous with the timescale over which the model begins to learn. As we will show below, $\tau_z \propto \|\mathbf{Z}^0\|_F / \tilde{\eta}$. Therefore in units of $\tilde{\eta}$, $\tau_z$ only depends on $\|\mathbf{Z}^0\|_F$ and not $\beta$.

To see this, note that at very short times it follows from Equation 5 that

$$\mathbf{Z}_i(\mathbf{x},t) - \mathbf{Z}_i(\mathbf{x},0) \approx \tilde{\eta}\sum_{j=1}^{K}(\hat{\Theta})_{ij}(\mathbf{x},\mathcal{X})(\mathcal{Y}_j - \sigma(\mathbf{Z}_j(\mathcal{X})))t + \mathcal{O}(t^2) \qquad (6)$$

It follows that we can define a timescale over which the logits (on the training set) change appreciably from their initial value as

$$\tau_z \equiv \frac{1}{\tilde{\eta}}\frac{\|\mathbf{Z}^0\|_F}{\|\hat{\Theta}(\mathcal{X},\mathcal{X})(\mathcal{Y} - \sigma(\mathbf{Z}^0))\|_F}. \qquad (7)$$

where the norms are once again taken across all classes as well as training points. This definition has the desired properties for $t \ll \tau_z$ and $t \gg \tau_z$.

In units of $\tilde{\eta}$, $\tau_z$ depends only on $\|\mathbf{Z}^0\|_F$, in two ways. The first is a linear scaling in $\|\mathbf{Z}^0\|_F$; the second comes from the contribution from the gradient $\|\hat{\Theta}(\mathcal{X},\mathcal{X})(\mathcal{Y} - \sigma(\mathbf{Z}(\mathcal{X},0)))\|_F$. As previously discussed, since $\sigma(\mathbf{Z}^0)$ saturates at small and large values of $\|\mathbf{Z}^0\|_F$, it follows that the gradient term will also saturate for large and small $\|\mathbf{Z}^0\|_F$, and the ratio of saturating values is some $\mathcal{O}(1)$ constant independent of $\|\mathbf{Z}^0\|_F$ and $\beta$.

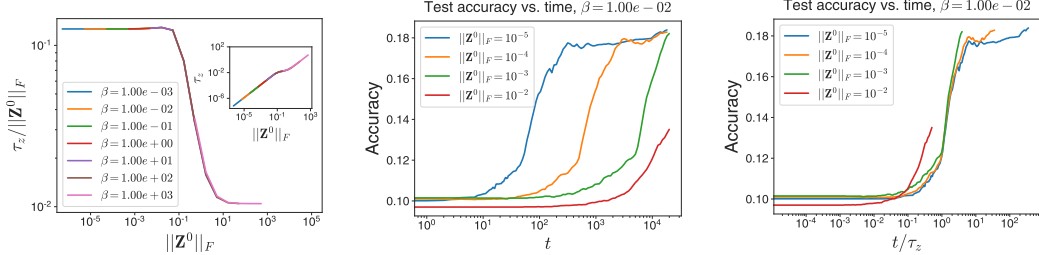

Figure 2: The timescale $\tau_z$ depends only on $\|\mathbf{Z}^0\|_F$ in units of $\tilde{\eta} = \beta^2\eta$ (left, inset). $\tau_z$ depends linearly on $\|\mathbf{Z}^0\|_F$, up to an $\mathcal{O}(1)$ coefficient which saturates at large and small $\|\mathbf{Z}^0\|_F$ (left, main). Accuracy increases more quickly for small initial $\|\mathbf{Z}^0\|_F$, though late time dynamics are similar (center). Rescaling time to $t/\tau_z$ causes early accuracy curves to collapse (right).

The quantitative and conceptual nature of $\tau_z$ can both be confirmed numerically. We compute $\tau_z$ explicitly as $\|[\mathbf{Z}_i(\mathbf{x},t) - \mathbf{Z}_i(\mathbf{x},0)]/t\|$ for short times (Figure 2, left). When plotted over a wide range of $\|\mathbf{Z}^0\|_F$ and $\beta$, the ratio $\tau_z/\|\mathbf{Z}^0\|_F$ (in rescaled time units) undergoes a saturating, $\mathcal{O}(1)$

variation from small to large $\|\mathbf{Z}^0\|_F$. The quantitative dependence of the transition on the NTK is confirmed in Appendix C. Additionally, for fixed $\beta$ and varying $\|\mathbf{Z}^0\|_F$, rescaling time by $1/\tau_z$ causes accuracy curves to collapse at early times (Figure 2, middle), even if they are very different at early times without the rescaling (right). We note here that the late time accuracy curves seem similar across $\|\mathbf{Z}^0\|_F$ without rescaling, a point which we will return to in Section 3.2.

## 2.4 NONLINEAR TIMESCALE

While linearized dynamics are useful to understand some features of learning, the best performing networks often reside in the nonlinear regime (Novak et al., 2019a). Here we define the nonlinear timescale, $\tau_{nl}$, corresponding to the time over which the network deviates appreciably from the linearized equations. We will show that $\tau_{nl} \propto \beta/\tilde{\eta}$. Therefore, in terms of $\beta$ and $\|\mathbf{Z}^0\|_F$, networks with small $\beta$ will access the nonlinear regime early in learning, while networks with large $\beta$ will be effectively linearized throughout training. We note that a similar point was raised in Chizat et al. (2019), primarily in the context of MSE loss.

We define $\tau_{nl}$ to be the timescale over which the change in $\hat{\Theta}_{\boldsymbol{\theta}}$ (which contributes to the second order term in Equation 6) can no longer be neglected. Examining the second time derivative of $\mathbf{Z}$, we have

$$\frac{d^2\mathbf{Z}_i}{dt^2} = \tilde{\eta}\sum_{j=1}^{K}\left(\underbrace{-(\hat{\Theta}_{\boldsymbol{\theta}})_{ij}(\mathbf{x},\mathcal{X})\frac{d}{dt}\sigma(\mathbf{Z}(\mathcal{X}))}_{\text{linearized dynamics}} + \underbrace{\frac{d}{dt}\left[(\hat{\Theta}_{\boldsymbol{\theta}})_{ij}(\mathbf{x},\mathcal{X})\right](\mathcal{Y}_j - \sigma(\mathbf{Z}_j(\mathcal{X})))}_{\text{nonlinearized dynamics} \equiv \ddot{\mathbf{Z}}_{nl}}\right) \quad (8)$$

The first term is the second derivative under a fixed kernel, while the second term is due to the change in the kernel (neglected in the linearized limit). A direct calculation shows that the second term, which we denote $\ddot{\mathbf{Z}}_{nl}$, can be written as

$$(\ddot{\mathbf{Z}}_{nl})_i = \beta^{-1}\tilde{\eta}^2\sum_{j=1}^{K}\sum_{k=1}^{K}(\mathcal{Y}_k(\mathcal{X}) - \sigma(\mathbf{Z}_k(\mathcal{X})))^{\mathrm{T}}\left(\frac{\partial\mathbf{z}_k(\mathcal{X})}{\partial\boldsymbol{\theta}}\cdot\frac{\partial}{\partial\boldsymbol{\theta}}[\hat{\Theta}_{\boldsymbol{\theta}}]_{ij}\right)(\mathcal{Y}_j(\mathcal{X}) - \sigma(\mathbf{Z}_j(\mathcal{X}))) \quad (9)$$

This gives us a *nonlinear timescale* $\tau_{nl}$ defined, at initialization, by $\tau_{nl} \equiv \|\dot{\mathbf{Z}}(\mathcal{X},0)\|_F/\|\ddot{\mathbf{Z}}_{nl}(\mathcal{X},0)\|_F$. We can interpret $\tau_{nl}$ as the time it takes for changes in the kernel to contribute to learning.

Though computing $\|\ddot{\mathbf{Z}}_{nl}(\mathcal{X},0)\|_F$ in exactly is analytically intractable, its basic scaling in terms of $\beta$ and $\|\mathbf{Z}^0\|_F$ (and therefore, that of $\tau_{nl}$) is computable. We first note the explicit $\beta^{-1}\tilde{\eta}^2$ dependence. The remaining terms are independent of $\beta$ and vary by at most $\mathcal{O}(1)$ with $\|\mathbf{Z}^0\|_F$; indeed as described above, $\|\mathcal{Y}(\mathcal{X}) - \sigma(\mathbf{Z}(\mathcal{X},0))\|_F$ saturates for large and small $\|\mathbf{Z}^0\|_F$. Morevoer, the derivative, $\frac{\partial\mathbf{z}(\mathcal{X},0)}{\partial\boldsymbol{\theta}}$, is the square root of the NTK and, at initialization, it is independent of $\|\mathbf{Z}^0\|_F$. Together with our analysis of $\tau_z$ we have that, up to some $O(1)$ dependence on $\|\mathbf{Z}^0\|_F$, $\tau_{nl} \propto \beta/\tilde{\eta}$. Therefore, the degree of nonlinearity early in learning is controlled via $\beta$ alone.

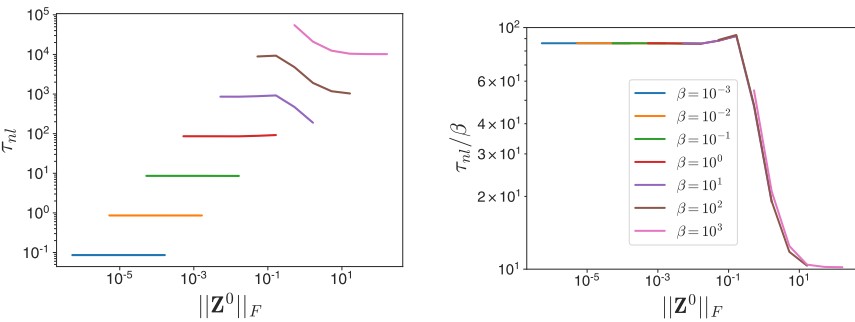

Figure 3: The time to deviation from linearized dynamics, $\tau_{nl}$, has large deviation over $\beta$ and $\|\mathbf{Z}^0\|_F$ (left), which can be largely explained by linear dependence on $\beta$ (right), in units of $\tilde{\eta} = \beta^2\eta$. There is an $O(1)$ dependence on $\|\mathbf{Z}^0\|_F$ which is consistent across varying $\beta$ for fixed $\|\mathbf{Z}^0\|_F$.

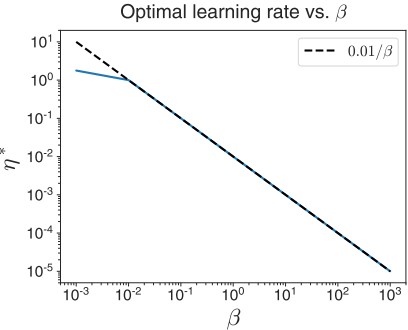

Figure 4: Optimal learning rate $\eta^*$ for WRN on CIFAR10 scales as $1/\beta$.

Once again we can confirm the quantitative and conceptual understanding of $\tau_{nl}$ numerically. Qualitatively, we see that for fixed $\|\mathbf{Z}^0\|_F$, models with smaller $\beta$ deviate sooner from the linearized dynamics when learning curves are plotted against $\tilde{\eta}t$ (Figure 1). We compute $\tau_{nl}$ explicitly by taking $\|[\mathbf{Z}(t) - \mathbf{Z}_{lin}(t)]/t^2\|$ as small times, where $\mathbf{Z}_{lin}(t)$ is the solution to Equation 5. $\tau_{nl}/\beta$ has an $O(1)$ dependence on $\|\mathbf{Z}^0\|_F$ only (Figure 3).

### 2.5 LEARNING RATES AND LEARNING SPEED

The timescales, $\tau_z$ and $\tau_{nl}$, can be combined to gain information about training with SGD. Indeed, the largest allowable learning rate is controlled by the curvature of the loss function Du et al. (2019); Allen-Zhu et al. (2019). A necessary condition for training is that the curvature be small compared with the step size, which happens when $\tau_{nl} \leq c$, where $c$ is an architecture-dependent constant of $O(1)$ which has not yet been calculated theoretically (see Lewkowycz et al. (2020) for empirical calculations of $c$). This predicts a maximum effective learning rate $\tilde{\eta}$ of $O(\beta)$ which in turn implies a raw learning rate of $O(\beta^{-1})$ (as $\eta = \tilde{\eta}/\beta^2$).

However, if $\tilde{\eta}$ is $O(\beta)$, then $\tau_z$ is $O(\beta^{-1})$. This means that, for early learning, networks with smaller $\beta$ will take more SGD steps to reach an appreciable change in logits. Therefore, we predict that networks with small $\beta$ take longer to train (which we see in practice as well).

## 3 EXPERIMENTAL RESULTS

### 3.1 OPTIMAL LEARNING RATE

We begin our empirical investigation by training wide resnets (Szegedy et al., 2016) without batch normalization on CIFAR10, as this architecture is well within the regime where our theory applies. In order to understand the effects of the different timescales on learning, we control $\beta$ and $\|\mathbf{Z}^0\|_F$ independently by using a correlated initialization strategy outlined in Appendix D.1.

Before considering model performance, it is first useful to understand the scaling of the learning rate with $\beta$. We define the *optimal* learning rate $\eta^*$ as the learning rate with the best generalization performance. To do this, we initialize networks with different $\beta$ and conduct learning rate sweeps for each $\beta$. The optimal learning rate $\eta^*$ has a clear $1/\beta$ dependence (Figure 4). This matches the prediction in Section 2.5, suggesting the maximum learning rate corresponds to the regime where the non-linear effects become important at the fastest rate for which training still converges. Again, as predicted, networks with smaller $\beta$ also learn more slowly in terms of number of SGD steps.

Thus, at small $\beta$ we expect learning to take place slowly and nonlinear effects to become important by the time the function has changed appreciably. At large $\beta$, by contrast, our results suggest that the network will have learned a significant amount before the dynamics become appreciably nonlinear.

## 3.2 PHASE PLANE

In the preceding discussion two quantities emerged that control the behavior of early-time dynamics: the inverse-temperature, $\beta$, and the rescaled logits $\|\mathbf{Z}^0\|_F$. In attempting to understand the behavior of real neural networks trained using softmax-cross-entropy loss, it therefore makes sense to try to reason about this behavior by considering neural networks that span the $\beta - \|\mathbf{Z}^0\|_F$ *phase plane*, the space of allowable pairs $(\beta, \|\mathbf{Z}^0\|_F)$. By construction, the phase plane is characterized by the timescales involved in early learning. To summarize, $\tau_z \sim \|\mathbf{Z}^0\|_F/\tilde{\eta}$ sets the timescale for early learning, with larger values of $\|\mathbf{Z}^0\|_F$ leading to longer time before significant accuracy gains are made (Section 2.3). Meanwhile, $\tau_{nl} \sim \beta/\tilde{\eta}$ controls the timescale for learning dynamics to leave the linearized regime - with small $\beta$ leading to immediate departures from linearity, while models with large $\beta$ may stay linearized throughout their learning trajectories (Section 2.4).

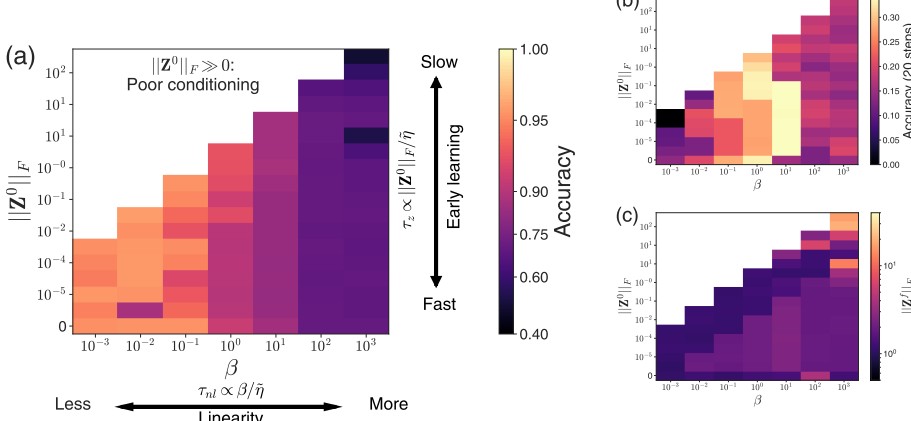

Figure 5: Properties of early learning dynamics, which affect generalization, can be determined by location in the $\beta$-$\|\mathbf{Z}^0\|_F$ phase plane (a). At optimal learning rate $\eta^*$, small $\beta$ and larger $\|\mathbf{Z}^0\|_F$ leads to slower early learning (b), and larger $\beta$ increases time before nonlinear dynamics contributes to learning. Large $\|\mathbf{Z}^0\|_F$ has poorly conditioned linearized dynamics. Generalization for a wide resnet trained on CIFAR10 is highly sensitive to $\beta$, and relatively insensitive to $\|\mathbf{Z}^0\|_F$ outside poor conditioning regime. Final logit variance is relatively insensitive to parameters (c).

In Figure 5 (a), we show a schematic of the phase plane. The colormap shows the test performance of a wide residual network (Zagoruyko and Komodakis, 2017), without batch normalization, trained on CIFAR10 in different parts of the phase plane. The value of $\beta$ makes a large difference in generalization, with optimal performance achieved at $\beta \approx 10^{-2}$. In general, larger $\beta$ performed worse than small $\beta$ as expected. Moreover, we observe similar generalization for all sufficiently large $\beta$; this is to be expected since models in this regime are close to their linearization throughout training (see Figure 1) and we expect the linearized models to have $\beta$-independent performance. Generalization was largely insensitive to $\|\mathbf{Z}^0\|_F$ so long as the network was sufficiently well-conditioned to be trainable. This suggests that long term learning is insensitive to $\tau_z$.

In Figure 5 (b), we plot the accuracy after 20 steps of optimization (with the optimal learning rate). For fixed $\|\mathbf{Z}^0\|_F$, the training speed was slow for the smallest $\beta$ and then became faster with increasing $\beta$. For fixed $\beta$ the training speed was fastest for small $\|\mathbf{Z}^0\|_F$ and slowed as $\|\mathbf{Z}^0\|_F$ increased. Both these phenomena were predicted by our theory and shows that both parameters are important in determining the early-time dynamics. However, we note that the relative accuracy across the phase plane at early times did not correlate with the performance at late times.

This highlights that differences in generalization are a dynamical phenomenon. Another indication of this fact is that at the end of training, at time $t_f$, the final training set logit values $\|\mathbf{Z}^f\|_F \equiv \|\mathbf{Z}(\mathcal{X}, t_f)\|_F$ tend towards 1 independent of the initial $\beta$ and $\|\mathbf{Z}^0\|_F$ (Figure 5, (c)). With the exception of the poorly-performing large $\|\mathbf{Z}^0\|_F$ regime, the different models reach similar levels of certainty by the end of training, despite having different generalization performances. Therefore generalization is not well correlated with the final model certainty (a typical motivation for tuning $\beta$).

### 3.3 ARCHITECTURE DEPENDENCE OF THE OPTIMAL $\beta$

Having demonstrated that $\beta$ controls the generalization performance of neural networks with softmax-cross-entropy loss, we now discuss the question of choosing the optimal $\beta$. Here we investigate this question through the lens of a number of different architectures. We find the optimal choice of $\beta$ to be strongly architecture dependent. Whether or not the optimal $\beta$ can be predicted analytically is an open question that we leave for future work. Nonetheless, we show that all architectures considered display optimal $\beta$ between approximately $10^{-2}$ and $10^1$. We observe that by taking the time to tune $\beta$ it is often the case that performance can be improved over the naive setting of $\beta = 1$.

#### 3.3.1 WIDE RESNET ON CIFAR10

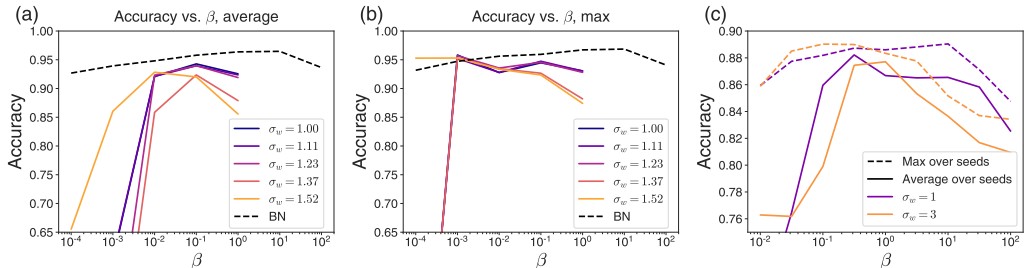

Figure 6: Dependence of test accuracy for various architectures with $\beta$ tuning. (a) For WRN with batchnorm, trained on CIFAR10, the optimal $\beta \approx 10$. Without batchnorm, the performance of the network can be nearly recovered with $\beta$-scaling alone with $\beta \approx 10^{-2}$. Even poorly conditioned networks (achieved by increasing weight scale $\sigma_w$) recover performance. (b) For $\beta < 10^{-2}$, learning is less stable, as evidenced by low average performance but high maximum performance (over 10 random seeds). (c) We see similar phenomenology on the IMDB sentiment analysis task trained with GRUs - where average-case best performance is near $\beta = 1$ but peak performance is at small $\beta$.

In Figure 6 (a) we show the accuracy against $\beta$ for several wide residual networks whose weights are drawn from normal distributions of different variances, $\sigma_w^2$, trained without batchnorm, as well as a network with $\sigma_w^2 = 1$ trained with batchnorm (averaged over 10 seeds). The best average performance is attained for $\beta < 1$, $\sigma_w = 1$ without batchnorm, and in particular networks with large $\sigma_w$ are dramatically improved with $\beta$ tuning. The network with batchnorm is better at all $\beta$, with optimal $\beta \approx 10$. However, we see that the best performing seed is often at a lower $\beta$ (Figure 6 (b)), with larger $\sigma_w$ networks competitive with $\sigma_w = 1$, and even with batchnorm at fixed $\beta$ (though batchnorm with $\beta = 10$ still performs the best). This suggests that small $\beta$ can improve best case performance, at the cost of stability. Our results emphasize the importance of tuning $\beta$, especially for models that have not otherwise been optimized.

#### 3.3.2 RESNET50 ON IMAGENET

Table 1: Accuracy on Imagenet dataset for ResNet-50. Tuning $\beta$ significantly improves accuracy.

| Method | Accuracy (%) |
|---|---|
| ResNet-50 (Ghiasi et al., 2018) | $76.51 \pm 0.07$ |
| ResNet-50 + Dropout (Ghiasi et al., 2018) | $76.80 \pm 0.04$ |
| ResNet-50 + Label Smoothing (Ghiasi et al., 2018) | $77.17 \pm 0.05$ |
| ResNet-50 + Temperature check ($\beta = 0.3$) | $77.37 \pm 0.02$ |

Motivated by our results on CIFAR10, we experimentally explored the effects of $\beta$ as a tunable hyperparameter for ResNet-50 trained on Imagenet. We follow the experimental protocol established by (Ghiasi et al., 2018). A key difference between this procedure and standard training is that we train for substantially longer: the number of training epochs is increased from 90 to 270. Ghiasi et al. (2018) found that this longer training regimen was beneficial when using additional regularization. Table 1 shows that scaling $\beta$ improves accuracy for ResNet-50 with batchnorm. However, we did not

find that using $\beta < 1$ was optimal for ResNet-50 without normalization. This further emphasizes the subtle architecture dependence that warrants further study.

### 3.3.3 GRUS ON IMDB SENTIMENT ANALYSIS

To further explore the architecture dependence of optimal $\beta$, we train GRUs (from Maheswaranathan et al. (2019)) whose weights are drawn from two different distributions on an IMDB sentiment analysis task that has been widely studied (Maas et al., 2011). We plot the results in Figure 6 (c) and observe that the results look qualitatively similar to the results on CIFAR10 without batch normalization. We observe a peak performance near $\beta \sim 1$ averaged over an ensemble of networks, but we observe that smaller $\beta$ can give better optimal performance at the expense of stability.

### 3.4 PROPOSED TUNING PROCEDURE

Our results suggest the following tuning procedure for networks trained with SGD/momentum:

- Train/tune a model as normal. Note the optimal learning rate $\eta_0$.
- For the best parameter set, sweep over $\beta \in [10^{-2}, 10^1]$, scaling learning rate as $\eta_0/\beta$.
- If best performing $\beta$ is at an endpoint of the range, continue tuning.

If there is sufficient compute, the $\beta$ search can instead be folded into the overall hyperparameter tuning.

We note that our observation of optimal $\beta < 1$ in classification settings stands in contrast to the observation of optimal $\beta > 1$ in the Bayesian inference setting (Wenzel et al., 2020). We speculate that the differences are related to the fact that in Bayesian inference, the uncertainty of estimates are important, and that the objective (learn an SDE which reproduces a target distribution) may have qualitatively different properties than classification tasks.

## 4 CONCLUSIONS

Our empirical results show that tuning $\beta$ can yield sometimes significant improvements to model performance. Perhaps most surprisingly, we observe gains on ImageNet even with the highly-optimized ResNet50 model. Our results on CIFAR10 suggest that the effect of $\beta$ may be even stronger in networks which are not yet highly-optimized, and results on IMDB show that this effect holds beyond the image classification setting. It is possible that even more gains can be made by more carefully tuning $\beta$ jointly with other hyperparameters, in particular the learning rate schedule and batch size.

One key lesson of our theoretical work is that properties of learning dynamics must be compared using the right units. For example, $\tau_{nl} \propto 1/\beta\eta$, which at first glance suggests that models with smaller $\beta$ will become nonlinear more slowly than their large $\beta$ counterparts. However, analyzing $\tau_{nl}$ with respect to the effective learning rate $\tilde{\eta} = \beta^2\eta$ yields $\tau_{nl} \propto \beta/\tilde{\eta}$. Thus we see that, in fact, networks with smaller $\beta$ tend to become more non-linearized before much learning has occurred, compared to networks with large $\beta$ which can remain in the linearized regime throughout training. Our numerical results confirm this intuition developed using the theoretical analysis.

As discussed above, our analysis suggests a range of good $\beta$, but does not predict the optimal value. Architecture-dependent, non-scaling multiplicative factors to key learning parameters have been observed in other contexts (Lewkowycz et al., 2020), and their numerical estimation is in general difficult. Extending the theoretical results to make predictions about these quantities is an interesting avenue for future work. Another area that warrants further study is the instability in training at small $\beta$.

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

# A   LINEARIZED LEARNING DYNAMICS

## A.1   FIXED POINTS

For the linearized learning dynamics, the trajectory $\mathbf{z}(\mathbf{x}, t)$ can be written in terms of the trajectories of the training set as

$$\mathbf{z}(\mathbf{x}, t) - \mathbf{z}(\mathbf{x}, 0) = \hat{\Theta}(\mathbf{x}, \mathcal{X})\hat{\Theta}^+(\mathcal{X}, \mathcal{X})(\mathbf{z}(\mathcal{X}, t) - \mathbf{z}(\mathcal{X}, 0)) \tag{10}$$

where $+$ is the pseudo-inverse. Therefore, if one can solve for $\mathbf{z}(\mathcal{X}, t)$, then in principle properties of generalization are computable.

However, in general Equation 3 does not admit an analytic solution even for fixed $\hat{\Theta}$, in contrast to the case of mean squared loss. It not even have an equilibrium - if the model can achieve perfect training accuracy, the logits will grow indefinitely. However, there is a guaranteed fixed point if the appropriate $L_2$ regularization is added to the training objective. Given a regularizer $\frac{1}{2}\lambda_{\boldsymbol{\theta}}\|\delta\boldsymbol{\theta}\|^2$ on the change in parameters $\delta\boldsymbol{\theta} = \boldsymbol{\theta}(t) - \boldsymbol{\theta}(0)$, the dynamics in the linearized regime are given by

$$\dot{\mathbf{z}}(\mathbf{x}) = \beta\eta\hat{\Theta}(\mathbf{x}, \mathcal{X})(\mathcal{Y}(\mathcal{X}) - \sigma(\beta\mathbf{z}(\mathcal{X}))) - \lambda_{\boldsymbol{\theta}}\delta\mathbf{z}(\mathbf{x}) \tag{11}$$

where the last term comes from the fact that $\frac{\partial \mathbf{z}}{\partial \boldsymbol{\theta}}\delta\boldsymbol{\theta} = \delta\mathbf{z}(\mathbf{x})$ in the linearized limit.

We can write down self-consistent equations for equilibria, which are approximately solvable in certain limits. For an arbitrary input $\mathbf{x}$, the equilibrium solution $\mathbf{z}^*(\mathbf{x})$ is

$$0 = \beta\hat{\Theta}(\mathbf{x}, \mathcal{X})(\mathcal{Y}(\mathcal{X}) - \sigma(\beta\mathbf{z}^*(\mathcal{X}))) - \lambda_{\boldsymbol{\theta}}\delta\mathbf{z}^*(\mathbf{x}) \tag{12}$$

This can be rewritten in terms of the training set as

$$\delta\mathbf{z}^*(\mathbf{x}) = \hat{\Theta}(\mathbf{x}, \mathcal{X})\hat{\Theta}^+(\mathcal{X}, \mathcal{X})\mathbf{z}^*(\mathcal{X}) \tag{13}$$

similar to kernel learning.

It remains then to solve for $\mathbf{z}^*(\mathcal{X})$. We have:

$$\delta\mathbf{z}^*(\mathcal{X}) = \frac{\beta}{\lambda_{\boldsymbol{\theta}}}\hat{\Theta}(\mathcal{X}, \mathcal{X})[\mathcal{Y}(\mathcal{X}) - \sigma(\beta\mathbf{z}^*(\mathcal{X}))] \tag{14}$$

We immediately note that the solution depends on the initialization. We assume $\mathbf{z}(\mathbf{x}, 0) = 0$, so $\delta\mathbf{z} = \mathbf{z}$ in order to simplify the analysis. The easiest case to analyze is when $\|\beta\mathbf{z}^*(\mathcal{X})\|_F \ll 1$. Then we have:

$$\mathbf{z}^*(\mathcal{X}) = \frac{\beta}{\lambda_{\boldsymbol{\theta}}}\hat{\Theta}(\mathcal{X}, \mathcal{X})\left[\mathcal{Y}(\mathcal{X}) - \frac{1}{K}(1 + \beta\mathbf{z}^*(\mathcal{X}))\right] \tag{15}$$

which gives us

$$\mathbf{z}^*(\mathcal{X}) = \frac{\beta}{\lambda_{\boldsymbol{\theta}}}\left[1 + \frac{\beta}{K\lambda_{\boldsymbol{\theta}}}\hat{\Theta}(\mathcal{X}, \mathcal{X})\right]^{-1}\hat{\Theta}(\mathcal{X}, \mathcal{X})(\mathcal{Y}(\mathcal{X}) - 1/K) \tag{16}$$

Therefore the self-consistency condition for this solution is $\|\frac{\beta}{\lambda_{\boldsymbol{\theta}}}\hat{\Theta}\|_F \ll 1$, which simplifies the solution to

$$\mathbf{z}^*(\mathcal{X}) = \frac{\beta}{\lambda_{\boldsymbol{\theta}}}\hat{\Theta}(\mathcal{X}, \mathcal{X})(\mathcal{Y}(\mathcal{X}) - 1/K) \tag{17}$$

This is equivalent to the solution after a single step of (full-batch) SGD with appropriate learning rate. We note that unlike linearized dynamics with $L_2$ loss and a full-rank kernel, there is no guarantee that the solution converges to 0 training error.

The other natural limit is $\|\beta\mathbf{z}^*(\mathcal{X})\|_2 \gg 1$. We focus on the 2 class case, in order to take advantage of the conserved quantity of learning with cross-entropy loss. We note that the vector on the right hand side of Equation 3 sums to 1 for every training point. Suppose at initialization, $\hat{\Theta}_{\boldsymbol{\theta}}$ has no logit-logit interactions, as is the case for most architectures in the infinite width limit with random initialization. More formally, we can write $\hat{\Theta}_{\boldsymbol{\theta}} = \mathbf{Id}_{K \times K} \otimes \hat{\Theta}_x$ where $\hat{\Theta}_x$ is $M \times M$. Then, the sum of the logits for any input $\mathbf{x}$ is conserved during linearized training, as we have:

$$\mathbf{1}^{\mathrm{T}}\dot{\mathbf{z}}(\mathbf{x}) = \eta\beta\mathbf{1}^{\mathrm{T}}\left[\mathbf{Id}_{K \times K} \otimes \hat{\Theta}_x\right](\mathcal{Y} - \sigma(\beta\mathbf{z}(\mathcal{X}))) \tag{18}$$

Multiplying the right hand side through, we get

$$\mathbf{1}^{\mathrm{T}}\dot{\mathbf{z}}(\mathbf{x}) = \eta\beta\hat{\Theta}_x\left[\mathbf{1}^{\mathrm{T}}(\mathcal{Y} - \sigma(\beta\mathbf{z}(\mathcal{X})))\right] = 0 \tag{19}$$

(Note that if $\hat{\Theta}_{\boldsymbol{\theta}}$ has explicit dependence on the logits, there still is a conserved quantity, which is more complicated to compute.)

Now we can analyze $\|\beta\mathbf{z}^*(\mathcal{X})\|_F \gg 1$. With two classes, and $\mathbf{z}(\mathcal{X}) = 0$ at initialization, we have $\mathbf{z}_1^* = -\mathbf{z}_2^*$. Therefore, without loss of generality, we focus on $\mathbf{z}_1^*$, the logit of the first class. In this limit, the leading order correction to the softmax is approximately:

$$\sigma(\beta\mathbf{z}_1^*) \approx \mathbf{1}_{\mathbf{z}_1^*>0} - \mathrm{sign}(\mathbf{z}_1^*)e^{-2\beta|\mathbf{z}_1^*|} \tag{20}$$

The self-consistency equation is then:

$$\mathbf{z}_1^*(\mathcal{X}) = \frac{\beta}{\lambda_{\boldsymbol{\theta}}}\hat{\Theta}(\mathcal{X},\mathcal{X})\left[\mathcal{Y}(\mathcal{X}) - \mathbf{1}_{\mathbf{z}_1^*>0} + \mathrm{sign}(\mathbf{z}_1^*)e^{-2\beta|\mathbf{z}_1^*|}\right] \tag{21}$$

The vector on the right hand side has entries that are $O(e^{-2\beta|\mathbf{z}_1^*|})$ for correct classifications, and $O(1)$ for incorrect ones. If we assume that the training error is 0, then we have:

$$\mathbf{z}_1^*(\mathcal{X}) = \frac{\beta}{\lambda_{\boldsymbol{\theta}}}\hat{\Theta}(\mathcal{X},\mathcal{X})\mathrm{sign}(\mathbf{z}_1^*)e^{-2\beta|\mathbf{z}_1^*|} \tag{22}$$

This is still non-trivial to solve, but we see that the self consistency condition is that $\ln(\beta\|\hat{\Theta}\|_F/\lambda_{\boldsymbol{\theta}}) \gg 1$.

Here also it may be difficult to train and generalize well. The individual elements of the right-hand-side vector are broadly distributed due to the exponential - so the outputs of the model are sensitive to/may only depend on a small number of datapoints. Even if the equilibrium solution has no training loss, generalization error may be high for the same reasons.

This suggests that even for NTK learning (with $L_2$ regularization), the scale of $\|\beta\mathbf{z}\|$ plays an important role in both good training accuracy and good generalization. In the NTK regime, there is one unique solution so (in the continuous time limit) the initialization doesn't matter; rather, the ratio of $\beta$ and $\lambda_{\boldsymbol{\theta}}$ (compared to the appropriate norm of $\hat{\Theta}$) needs to be balanced to prevent falling into the small $\beta\mathbf{z}$ regime (where training error might be large) or the large $\beta\mathbf{z}$ regime (where a few datapoints might dominate and reduce generalization).

## A.2 Dynamics near equilibrium

The dynamics near the equilibrium can be analyzed by expanding around the fixed point equation. We focus on the dynamics on the training set. The dynamics of the difference $\tilde{\mathbf{z}}(\mathcal{X}) = \mathbf{z}(\mathcal{X}) - \mathbf{z}^*(\mathcal{X})$ for small perturbations is given by

$$\dot{\tilde{\mathbf{z}}}(\mathcal{X}) = -\eta\left[\beta^2[\mathbf{Id}_{\mathbf{z}} \otimes \hat{\Theta}(\mathcal{X},\mathcal{X})]\boldsymbol{\sigma}_z(\beta\mathbf{z}^*(\mathcal{X})) + \lambda_{\boldsymbol{\theta}}\right]\tilde{\mathbf{z}}(\mathcal{X}) \tag{23}$$

where $\boldsymbol{\sigma}_z$ is the derivative of the softmax matrix

$$\boldsymbol{\sigma}_z(\mathbf{z}) \equiv \frac{\partial\sigma(\mathbf{z})}{\partial\mathbf{z}'} = \mathbf{diag}(\sigma(\mathbf{z})) - \sigma(\mathbf{z})\sigma(\mathbf{z}')^{\mathrm{T}} \tag{24}$$

We can perform some analysis in the large and small $\beta$ cases (once again ignoring $\lambda_z$). For small $\|\beta\mathbf{z}^*(\mathcal{X})\|_F$, we have $\|\frac{\beta}{\lambda_{\boldsymbol{\theta}}}\hat{\Theta}\|_F \ll 1$ which leads to:

$$\boldsymbol{\sigma}_z(\beta\mathbf{z}^*(\mathcal{X})) = (1/K - \mathbf{11}^{\mathrm{T}}/K^2) + O(\beta\hat{\Theta}) \tag{25}$$

This matrix has $K - 1$ eigenvalues with value $1/K$, and one zero eigenvalue (corresponding to the conservation of probability). Therefore $\|\beta^2[\mathbf{Id}_{\mathbf{z}} \otimes \hat{\Theta}(\mathcal{X},\mathcal{X})]\boldsymbol{\sigma}_z(\beta\mathbf{z}^*(\mathcal{X}))\|_F \ll \lambda_{\boldsymbol{\theta}}$, and the well-conditioned regularizer dominates the approach to equilibrium.

In the large $\beta$ case ($\ln(\beta\|\hat{\Theta}\|/\lambda_{\boldsymbol{\theta}}) \gg 1$), the values of $\sigma(\beta\mathbf{z}(\mathcal{X}))$ are exponentially close to $0$ ($K - 1$ values) or $1$ (the value corresponding to the largest logit). This means that $\boldsymbol{\sigma}_z(\beta\mathbf{z}(\mathcal{X}))$ has exponentially small values in $\|\beta\mathbf{z}(\mathcal{X})\|_F$ - if any one of $\sigma(\beta\mathbf{z}_i(\mathcal{X}))$ and $\sigma(\beta\mathbf{z}_j(\mathcal{X}))$ is exponentially

small, the corresponding element of $\boldsymbol{\sigma}_z(\beta\mathbf{z}(\mathcal{X}))$ is as well; for the largest logit $i$ the diagonal is $\sigma(\beta\mathbf{z}_i(\mathcal{X}))(1 - \sigma(\beta\mathbf{z}_i(\mathcal{X})))$ which is also exponentially small.

From Equation 22, we have $\lambda_{\boldsymbol{\theta}} \ll \beta^2 e^{2\beta|\mathbf{z}_1^*|}$; therefore, though the $\boldsymbol{\sigma}_z$ term of $\mathbf{H}$ is exponentially small, it dominates the linearized dynamics near the fixed point, and the approach to equilibrium is slow. We will analyze the conditioning of the dynamics in the remainder of this section.

## A.3 CONDITIONING OF DYNAMICS

Understanding the conditioning of the linearized dynamics requires understanding the spectrum of the Hessian matrix $\mathbf{H} = \left(\mathbf{Id}_{\mathbf{z}} \otimes \hat{\Theta}(\mathcal{X}, \mathcal{X})\right) \boldsymbol{\sigma}_z(\beta\mathbf{z}^*(\mathcal{X}))$. In the limit of large model size, the first factor is block-diagonal with training set by training set blocks (no logit-logit interactions), and the second term is block-diagonal with $K \times K$ blocks (no datapoint-datapoint interactions).

We will use the following lemma to get bounds on the conditioning:

**Lemma:** Let $\mathbf{M} = \mathbf{AB}$ be a matrix that is the product of two matrices. The condition number $\kappa(\mathbf{M}) \equiv \frac{\lambda_{\mathbf{M},\max}}{\lambda_{\mathbf{M},\min}}$ has bound

$$\kappa(\mathbf{B})/\kappa(\mathbf{A}) \leq \kappa(\mathbf{M}) \leq \kappa(\mathbf{A})\kappa(\mathbf{B}) \tag{26}$$

**Proof:** Consider the vector $\mathbf{v}$ that is the eigenvector of $\mathbf{B}$ associated with $\lambda_{\mathbf{B},\min}$. Note that $||\mathbf{Av}||/||\mathbf{v}|| \leq \lambda_{\mathbf{A},\max}$. Analogously, for $\mathbf{w}$, the eigenvector associated with $\lambda_{\mathbf{B},\max}$, $||\mathbf{Aw}||/||\mathbf{w}|| \geq \lambda_{\mathbf{A},\min}$. This gives us the two bounds:

$$\lambda_{\mathbf{M},\min} \leq \lambda_{\mathbf{A},\max}\lambda_{\mathbf{B},\min}, \; \lambda_{\mathbf{M},\max} \geq \lambda_{\mathbf{A},\min}\lambda_{\mathbf{B},\max} \tag{27}$$

This means that the condition number $\kappa(\mathbf{H}) \equiv \frac{\lambda_{\mathbf{M},\max}}{\lambda_{\mathbf{M},\min}}$ is bounded by

$$\kappa(\mathbf{M}) \geq \frac{\lambda_{\mathbf{A},\max}\lambda_{\mathbf{B},\min}}{\lambda_{\mathbf{A},\min}\lambda_{\mathbf{B},\max}} = \kappa(\mathbf{B})/\kappa(\mathbf{A}) \tag{28}$$

In total, we have the bound of Equation 26, where the upper bound is trivial to prove. $\square$

In particular, this means that a poorly conditioned $\boldsymbol{\sigma}_z(\beta\mathbf{z}^*(\mathcal{X}))$ will lead to poor conditioning of the linearized dynamics if the NTK $\hat{\Theta}(\mathcal{X}, \mathcal{X})$ is (relatively) well conditioned. This bound will be important in establishing the poor conditioning of the linearized dynamics for the large logit regime $||\beta\mathbf{z}|| \gg 1$.

### A.3.1 SMALL LOGIT CONDITIONING

For $||\beta\mathbf{z}^*(\mathcal{X})||_F \ll 1$, the Hessian $\mathbf{H}$ is

$$\mathbf{H} = \frac{1}{K}\left(1 - \frac{1}{K}\mathbf{1}\mathbf{1}^{\mathrm{T}}\right) \otimes \hat{\Theta}(\mathcal{X}, \mathcal{X}) \tag{29}$$

Since $\mathbf{H}$ is the Kroenecker product of two matrices, the condition numbers multiply, and we have

$$\kappa(\mathbf{H}) = \kappa(\hat{\Theta}) \tag{30}$$

which is well-conditioned so long as the NTK is. Regardless, the well-conditioned regularization due to $\lambda_{\boldsymbol{\theta}}$ dominates the approach to equilibrium.

### A.3.2 LARGE LOGIT CONDITIONING

Now consider $||\beta\mathbf{z}^*(\mathcal{X})||_F \gg 1$. Here we will show that the linearized dynamics is poorly conditioned, and that $\kappa(\mathbf{H})$ is exponentially large in $\beta$.

We first try to understand $\boldsymbol{\sigma}_z(\beta\mathbf{z}^*(\mathbf{x}))$ for an individual $\mathbf{x} \in \mathcal{X}$. To 0th order (in an as-of-yet-undefined expansion), $\boldsymbol{\sigma}_z$ is zero - at large temperature the softmax returns either 0 or 1, which by Equation 24 gives 0 in all entries. The size of the corrections end up being exponentially dependent on $||\beta\mathbf{z}^*||_F$; the entries will have broad, log-normal type distributions with magnitudes which scale

as $\exp(-\beta|\mathbf{z}_1^*|)$. There will be two scaling regimes one with a small number of labels in the sense $\sqrt{\beta} \gg \sqrt{\ln(K)}$, where the largest logit dominates the statistics, and one where the number of labels is large (and the central limit theorem applies to the partition function). In both cases, however, there is still exponential dependence on $\beta$; we will focus on the first which is easier to analyze and more realistic (e.g. for $10^6$ labels "large" $\beta$ is only $\sim 15$).

Let $z_1$ be the largest of $K$ logits, $z_2$ the second largest, and so on. Then using Equation 24 we have:

$$(\boldsymbol{\sigma}_z)_{i1} = -e^{-\beta(z_1-z_i)} \tag{31}$$

for $i \neq 1$,

$$(\boldsymbol{\sigma}_z)_{ij} = \delta_{ij}e^{-\beta(z_1-z_i)} - e^{-\beta(2z_1-z_i-z_j)} \tag{32}$$

for $i \neq j$ and

$$(\boldsymbol{\sigma}_z)_{11} = e^{-\beta(z_1-z_2)} \tag{33}$$

The eigenvectors and eigenvalues can be approximately computed as:

$$(\mathbf{v}_1)_1 = \frac{1}{\sqrt{2}}, \ (\mathbf{v}_1)_2 = -\frac{1}{\sqrt{2}}, \ \lambda_1 = 2e^{-\beta(z_1-z_2)} \tag{34}$$

$$(\mathbf{v}_2)_1 = \frac{1}{\sqrt{2}}, \ (\mathbf{v}_2)_2 = \frac{1}{\sqrt{2}}, \ \lambda_2 = -\frac{1}{2}e^{-2\beta(z_1-z_2)} \tag{35}$$

and for $i > 2$,

$$(\mathbf{v}_i)_i = 1, \ (\mathbf{v}_i)_1 = e^{-\beta(z_2-z_i)}, \ \lambda_i = e^{-\beta(z_1-z_i)} \tag{36}$$

with all non-explicit eigenvector components 0. This expansion is valid provided that $\beta/K \gg 1$ (so that $e^{\beta(z_1-z_i)} \gg e^{\beta(z_1-z_{i+1})}$).

Therefore the spectrum of any individual block $\boldsymbol{\sigma}_z(\beta\mathbf{z}(x))$ is exponentially small in $\beta$. Using the bound in the Lemma, we have:

$$\kappa(\mathbf{H}) \geq e^{O(\beta|\mathbf{z}_1^*|)}/\kappa(\hat{\Theta}(\mathcal{X}, \mathcal{X})) \tag{37}$$

This is a very loose bound, as it assumes that the largest eigendirections of $\boldsymbol{\sigma}_z$ are aligned with the smallest eigendirections of $\mathbf{Id}_\mathbf{z} \otimes \hat{\Theta}$, and vice versa. It is possible $\kappa(\mathbf{H})$ is closer in magnitude to the upper bound $e^{\beta(z_2-z_K)}\kappa(\hat{\Theta}(\mathcal{X}, \mathcal{X}))$.

Regardless, $\kappa(\mathbf{H})$ is exponentially large in $\beta$ - meaning that the conditioning is exponentially poor for large $\|\beta\mathbf{z}^*\|_F$.

# B  SGD AND MOMENTUM RESCALINGS

## B.1  DISCRETE EQUATIONS

Consider full-batch SGD training. The update equations for the parameters $\boldsymbol{\theta}$ are:

$$\boldsymbol{\theta}_{t+1} = \boldsymbol{\theta}_t - \eta\nabla_{\boldsymbol{\theta}}\mathcal{L} \tag{38}$$

We will denote $\mathbf{g}_t \equiv \nabla_{\boldsymbol{\theta}}\mathcal{L}$ for ease of notation.

Training with momentum, the equations of motion are given by:

$$\mathbf{v}_{t+1} = (1-\gamma)\mathbf{v}_t - \mathbf{g}_t \tag{39}$$

$$\boldsymbol{\theta}_{t+1} = \boldsymbol{\theta}_t + \eta\mathbf{v}_{t+1} \tag{40}$$

where $\gamma \in [0, 1]$.

One key point to consider later will be the relative magnitude $\Delta_{\boldsymbol{\theta}}$ of updates to the parameters. For SGD, the magnitude of updates is $\eta\|\mathbf{g}\|$. For momentum with slowly-varying gradients the magnitude is $\eta\|\mathbf{g}\|/\gamma$.

## B.2 CONTINUOUS TIME EQUATIONS

We can write down the continuous time version of the learning dynamics as follows. For SGD, for small learning rates we have:

$$\frac{d\boldsymbol{\theta}}{dt} = -\eta \mathbf{g} \tag{41}$$

For the momentum equations we have

$$\frac{d\mathbf{v}}{dt} = -\gamma \mathbf{v} - \mathbf{g} \tag{42}$$

$$\frac{d\boldsymbol{\theta}}{dt} = \eta \mathbf{v} \tag{43}$$

From these equations, we can see that in the continuous time limit, there are coordinate transformations which can be used to cause sets of trajectories with different parameters to collapse to a single trajectory. SGD is the simplest, where rescaling time to $\tau \equiv \eta t$ causes learning curves to be identical for all learning rates.

For momentum, instead of a single universal learning curve, there is a one-parameter family of curves controlled by the ratio $T_{mom} \equiv \eta/\gamma^2$. Consider rescaling time to $\tau = at$ and $\boldsymbol{\nu} = b\mathbf{v}$, where $a$ and $b$ will be chosen to put the equations in a canonical form. In our new coordinates, we have

$$\frac{d\boldsymbol{\nu}}{d\tau} = -(\gamma/a)\boldsymbol{\nu} - (b/a)\mathbf{g} \tag{44}$$

$$\frac{d\boldsymbol{\theta}}{d\tau} = \eta \boldsymbol{\nu}/(ab) \tag{45}$$

The canonical form we choose is

$$\frac{d\boldsymbol{\nu}}{d\tau} = -\lambda \boldsymbol{\nu} - \mathbf{g} \tag{46}$$

$$\frac{d\boldsymbol{\theta}}{d\tau} = \boldsymbol{\nu} \tag{47}$$

From which we arrive at $a = b = \sqrt{\eta}$, which gives us $\lambda = \gamma/\sqrt{\eta}$.

Note that this is not a unique canonical form; for example, if we fix a coefficient of $-1$ on $\boldsymbol{\nu}$, we end up with

$$\frac{d\boldsymbol{\nu}}{d\tau} = -\boldsymbol{\nu} - (\eta/\gamma^2)\mathbf{g} \tag{48}$$

$$\frac{d\boldsymbol{\theta}}{d\tau} = \boldsymbol{\nu} \tag{49}$$

with $a = \gamma$. This is a different time rescaling, but still controlled by $T_{mom}$.

Working in the canonical form of Equations 46 and 47, we can analyze the dynamics. One immediate question is the difference between $\lambda \ll 1$ and $\lambda \gg 1$. We note that the integral equation

$$\boldsymbol{\nu}(\tau) = \boldsymbol{\nu}(0) + \int_0^\tau e^{-\lambda(\tau-\tau')}\mathbf{g}(\tau')d\tau' \tag{50}$$

solves the differential equation for $\boldsymbol{\nu}$. Therefore, for $\lambda \gg 1$, $\boldsymbol{\nu}(t)$ only depends on the current value $\mathbf{g}(t)$ and we have $\boldsymbol{\nu}(\tau) \approx \mathbf{g}(\tau)/\lambda$. Therefore, we have, approximately:

$$\frac{d\boldsymbol{\theta}}{d\tau} \approx \frac{1}{\lambda}\mathbf{g} \tag{51}$$

This means that for large $\lambda$ all the curves will approximately collapse, with timescale given by $\sqrt{\eta}\lambda^{-1} = \gamma\eta$ (dynamics similar to SGD).

For $\lambda \ll 1$, the momentum is essentially the integrated gradient across all time. If $\boldsymbol{\nu}(0) = 0$, then we have

$$\frac{d\boldsymbol{\theta}}{d\tau} \approx \int_0^\tau \mathbf{g}(\tau')d\tau' \tag{52}$$

In this limit, $\boldsymbol{\theta}(\tau)$ is the double integral of the gradient with respect to time.

Given the form of the controlling parameter $T_{mom}$, we can choose to parameterize $\gamma = \tilde{\gamma}\sqrt{\eta}$. Under this parameterization, we have $T_{mom} = \tilde{\gamma}^2$. The dynamical equations then become:

$$\frac{d\boldsymbol{\nu}}{d\tau} = -\tilde{\gamma}\boldsymbol{\nu} - \mathbf{g} \tag{53}$$

$$\frac{d\boldsymbol{\theta}}{d\tau} = \boldsymbol{\nu} \tag{54}$$

which automatically removes explicit dependence on $\eta$.

One particular regime of interest is the early-time dynamics, starting from $\boldsymbol{\nu}(0) = 0$. Integrating directly, we have:

$$\boldsymbol{\theta}(\tau) = -\frac{1}{2}\mathbf{g}\tau^2 + \frac{1}{6}\tilde{\gamma}\mathbf{g}\tau^3 + \ldots \tag{55}$$

This means that $\tau$ alone is the correct timescale for early learning, at least until $\tau\tilde{\gamma} \sim 1$ - which in the original parameters corresponds to $t \sim 1/\gamma$ (the time it takes for the momentum to be first "fully integrated").

### B.3 DETAILED ANALYSIS OF MOMENTUM TIMESCALES

One important subtlety is that $1$ is not the correct value to compare $\lambda$ to. The real timescale involved is the one over which $\mathbf{g}$ changes significantly. We can approximate this in the following way. Suppose that there is some relative change $\frac{\Delta_{\boldsymbol{\theta}}}{||\boldsymbol{\theta}||} \sim c$ of the parameters that leads to an appreciable relative change in $\mathbf{g}$. Then the timescale over which $\boldsymbol{\theta}$ changes by that amount is the one we must compare $\lambda$ to.

We can compute that timescale in the following way. We assume $\mathbf{g}$ fixed for what follows. Therefore, Equation 51 approximately holds. The timescale $\tau_c$ of the change is then given by:

$$\frac{\Delta_{\boldsymbol{\theta}}}{||\boldsymbol{\theta}||} = \frac{1}{\lambda}\frac{||\mathbf{g}||}{||\boldsymbol{\theta}||}\tau_c \sim c \tag{56}$$

which gives

$$\tau_c \sim c\lambda||\boldsymbol{\theta}||/||\mathbf{g}|| \tag{57}$$

In particular, this means that the approximation is good when $\lambda\tau_c \gg 1$, which gives $\gamma^2/\eta \gg \frac{||\mathbf{g}||}{||\boldsymbol{\theta}||}$ - the former being a function of the dynamical parameters, the latter being a function of the geometry of $\mathcal{L}$ with respect to $\boldsymbol{\theta}$.

One consequence of this analysis is that if the $||\boldsymbol{\theta}||$ remains roughly constant, for fixed $\eta$ and $\gamma$, late in learning when the gradients become small the dynamics shifts into the regime where $\lambda$ is large, and we effectively have SGD.

### B.4 CONNECTING DISCRETE AND CONTINUOUS TIME

One use for the form of the continuous time rescalings is to use them to compare learning curves for the actual discrete optimization that is performed with different learning rates. For small learning rates, the curves are expected to collapse, while for larger learning rates the deviations from the continuous expectation can be informative.

With momentum, we only have perfect collapse when $\gamma$ and $\eta$ are scaled together. However, one typical use case for momentum is to fix the parameter $\gamma$, and sweep through different learning rates. With this setup, if $\mathbf{g}$ is changing slowly compared to $\gamma$ (more precisely, $\gamma^2/\eta \gg ||\mathbf{g}||/||\boldsymbol{\theta}||$), as may be the case at later training times, the change in parameters from a single step is $\Delta_{\boldsymbol{\theta}} \sim (\eta/\gamma))||\mathbf{g}||$ and the rescaling of taking $t$ to $\eta t$ (as for SGD) collapses the dynamics. Therefore given a collection of varying $\eta$, but fixed $\gamma$ curves, it is possible to get intermediate and late time dynamics on the same scale.

However, at early times, while the momentum is still getting "up to speed" (i.e. in the first $1/\gamma$ steps), the appropriate timescale is $\eta^{-1/2}$. Therefore, in order to get learning curves to collapse across

different $\eta$ at early times, we need to rescale $\gamma$ with $\eta$ as implied by Equations 46 and 47. Namely, one must fix $\tilde{\gamma}$ and rescale $\gamma = \tilde{\gamma}\sqrt{\eta}$. We note that, since $\gamma < 1$, this gives us a restriction $\eta < \tilde{\gamma}^{-2}$ for the maximum learning that can be supported by the rescaled momentum parameter.

### B.5 Momentum equations with softmax-cross-entropy learning

For cross-entropy learning with softmax inputs $\beta \mathbf{z}$, all the scales acquire dependence on $\beta$. If we define $\ell_z \equiv \left|\left|\frac{\partial \mathcal{L}}{\partial \beta \mathbf{z}}\right|\right|$ and $g_z \equiv \left|\left|\frac{\partial \mathbf{z}}{\partial \boldsymbol{\theta}}\right|\right|_F$, then we have, approximately, $||\mathbf{g}|| \approx \beta \ell_z g_z$.

Consider the goal of obtaining identical early-time learning curves for different values of $\beta$. (The curves are only globally consistent across $\beta$ in the linearized regime.) In order to get learning curves to collapse, we want $\frac{d\mathcal{L}}{d\tau}$ to be independent of $\beta$ in the rescaled time units. We note that the change in the loss function $\Delta_{\mathcal{L}}$ from a single step of SGD goes as

$$\Delta_{\mathcal{L}} \sim \eta \beta^2 \ell_z^2 g_z^2 \tag{58}$$

This suggests that one way to collapse learning curves is to plot them against the rescaled learning rate $\tilde{\eta}t$, where $\tilde{\eta} = \eta \beta^2$. While hyperparameter tuning across $\beta$, one could use $\eta = \tilde{\eta}/\beta^2$, sweeping over $\tilde{\eta}$ in order to easily obtain comparable learning curves.

However, a better goal for a learning rate rescaling is to try and stay within the continuous time limit - that is, to control the change in parameters $\Delta_{\boldsymbol{\theta}}$ for a single step to be small across $\beta$. We have

$$\Delta_{\boldsymbol{\theta}} \sim \eta \beta \ell_z g_z \tag{59}$$

which suggests that maximum allowable learning rates will scale as $1/\beta$. This suggests setting $\eta = \hat{\eta}\beta^{-1}$, and rescaling time as $\hat{\eta}\beta$ in order to best explore the continuous learning dynamics.

We can perform a similar analysis for the momentum optimizers. We begin by analyzing the continuous time equations for the dynamics of the loss. Starting with the rescalings from Equations 53 and 54 we have

$$\frac{d\boldsymbol{\nu}}{d\tau} = -\tilde{\gamma}\boldsymbol{\nu} - \beta \mathbf{g} \tag{60}$$

$$\frac{d\boldsymbol{\theta}}{d\tau} = \boldsymbol{\nu} \tag{61}$$

$$\frac{d\mathcal{L}}{d\tau} = \beta \mathbf{g} \cdot \boldsymbol{\nu} \tag{62}$$

where $\mathbf{g} = \frac{\partial \mathcal{L}}{\partial \beta \mathbf{z}} \frac{\partial \mathbf{z}}{\partial \boldsymbol{\theta}}$. Rescaling $\tau$ by $\beta$ gets us:

$$\frac{d\boldsymbol{\nu}}{d\beta\tau} = -\frac{\tilde{\gamma}}{\beta}\boldsymbol{\nu} - \mathbf{g} \tag{63}$$

$$\frac{d\boldsymbol{\theta}}{d\beta\tau} = \frac{1}{\beta}\boldsymbol{\nu} \tag{64}$$

$$\frac{d\mathcal{L}}{d\beta\tau} = \mathbf{g} \cdot \boldsymbol{\nu} \tag{65}$$

This rescaling causes a collapse of the trajectories of the $\mathcal{L}$ at early times if $\tilde{\gamma}/\beta$ is constant for varying $\beta$.

One scheme to arrive at the final canonical form, across $\beta$, is by the following definitions of $\eta$, $\gamma$, and $\tau$:

- $\eta = \tilde{\eta}\beta^{-2}$
- $\gamma \equiv \beta\sqrt{\eta}\tilde{\gamma} = \sqrt{\tilde{\eta}}\tilde{\gamma}$
- $\tau \equiv \beta\sqrt{\eta}t = \sqrt{\tilde{\eta}}t$

where curves with fixed $\tilde{\gamma}$ will collapse. The latter two equations are similar to before, except with $\eta$ replaced with $\tilde{\eta}$. The dynamical equations are then:

$$\frac{d\boldsymbol{\nu}}{d\tau} = -\tilde{\gamma}\boldsymbol{\nu} - \mathbf{g} \tag{66}$$

$$\frac{d\boldsymbol{\theta}}{d\tau} = \frac{1}{\beta}\boldsymbol{\nu} \tag{67}$$

$$\frac{d\mathcal{L}}{d\tau} = \mathbf{g} \cdot \boldsymbol{\nu} \tag{68}$$

The change in parameters from a single step (assuming constant $\mathbf{g}$ and saturation) is

$$\Delta_{\boldsymbol{\theta}} = \frac{\|\mathbf{g}\|}{\tilde{\gamma}\beta}\sqrt{\hat{\eta}} \tag{69}$$

If we instead want the change in parameters from a single step to be invariant of $\beta$ so the continuous time approximation holds, while maintaining collapse of trajectories, we first note that

$$\Delta_{\boldsymbol{\theta}} \sim \frac{\sqrt{\eta}}{\tilde{\gamma}}\beta\ell_z g_z \tag{70}$$

from a single step of the momentum optimizer. To keep $\Delta_{\boldsymbol{\theta}}$ invariant of $\beta$, we can set:

- $\eta = \hat{\eta}\beta^{-1}$
- $\gamma \equiv \beta\sqrt{\eta}\tilde{\gamma} = \sqrt{\hat{\eta}}\tilde{\gamma} = \beta^{1/2}\sqrt{\hat{\eta}}\tilde{\gamma}$
- $\tau \equiv \beta\sqrt{\eta}t = \beta^{1/2}\sqrt{\hat{\eta}}t$

Note that the relationship between $\gamma$ and $\tilde{\gamma}$ is the same in both schemes when measured with respect to the raw learning rate $\eta$.

## C  SOFTMAX-CROSS-ENTROPY GRADIENT MAGNITUDE

### C.1  MAGNITUDE OF GRADIENTS IN FULLY-CONNECTED NETWORKS

The value of $\tau_z$ has nontrivial (but bounded) dependence on $\|\mathbf{Z}^0\|_F$ via the $\|\hat{\Theta}(\mathcal{Y} - \sigma(\mathbf{Z}^0(\mathcal{X})))\|_F$ term in Equation 7. We can confirm the dependence for highly overparameterized models by using the theoretical $\hat{\Theta}$. In particular, for wide neural networks, the tangent kernel is block-diagonal in the logits, and easily computable.

The numerically computed $\tau_z/\|\mathbf{Z}^0\|_F$ correlates well with $\|\hat{\Theta}(\mathcal{Y} - \sigma(\mathbf{Z}^0(\mathcal{X})))\|_F^{-1}$ for wide (2000 hidden units/layer) fully connected networks (Figure 7). The ratio depends on details like the nonlinearities in the network; for example, Relu units tend to have a larger ratio than Erf units (left and middle). The ratio also depends on the properties of the dataset. For example, the ratio increases on CIFAR10 when the training labels are randomly shuffled (right).

Therefore in general the ratio of $\tau_z/\|\mathbf{Z}^0\|_F$ at large and small $\|\mathbf{Z}^0\|_F$ depends subtly on the relationship between the NTK and the properties of the data distribution. A full analysis of this relationship is beyond the scope of this work. The exact form of the transition is likely even more sensitive to these properties and is therefore harder to analyze than the ratio alone.

## D  EXPERIMENTAL DETAILS

### D.1  CORRELATED INITIALIZATION

In order to avoid confounding effects of changing $\beta$ and $\|\mathbf{Z}^0\|_F$ with changes to $\hat{\Theta}$, we use a *correlated initialization* strategy (similar to (Chizat et al., 2019)) which fixes $\hat{\Theta}$ while allowing for independent variation of $\beta$ and $\|\mathbf{Z}^0\|_F$. Given a network with final hidden layer $\mathbf{h}(\mathbf{x}, \boldsymbol{\theta})$ and output weights $\mathbf{W}_O$, we define a combined network $\mathbf{z}_c(\mathbf{x}, \tilde{\boldsymbol{\theta}})$ explicitly as

$$\mathbf{z}_c(\mathbf{x}, \mathbf{W}_{O,1}, \mathbf{W}_{O,2}, \boldsymbol{\theta}_1, \boldsymbol{\theta}_2) = \mathbf{W}_{O,1}\mathbf{h}(\mathbf{x}, \boldsymbol{\theta}_1) + \mathbf{W}_{O,2}\mathbf{h}(\mathbf{x}, \boldsymbol{\theta}_2) \tag{71}$$

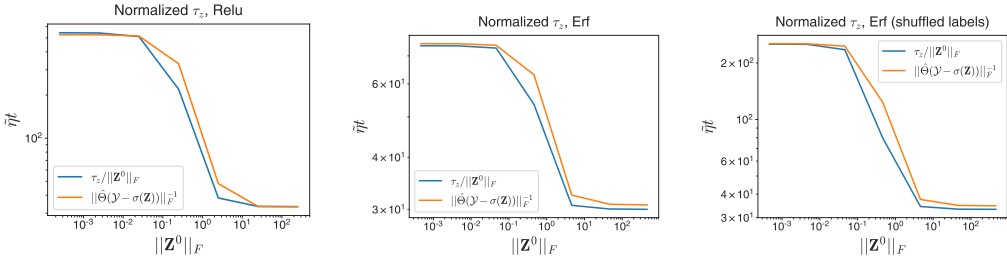

Figure 7: $\tau_z/\|\mathbf{Z}^0\|_F$ is highly correlated with $\|\hat{\Theta}(\mathcal{Y} - \sigma(\mathbf{Z}^0))\|^{-1}$, with $\hat{\Theta}$ computed in the infinite width limit (in units of effective learning rate $\tilde{\eta} = \beta^2\eta$). Ratio between normalized timescales at large and small $\|\mathbf{Z}^0\|_F$ depends on nonlinearity (left and middle), as well as training set statistics (right, CIFAR10 with shuffled labels).

where, at initialization, $\boldsymbol{\theta}_1 = \boldsymbol{\theta}_2$, and the elements of $\mathbf{W}_{O,a}$ have statistics

$$\mathrm{E}[(\mathbf{W}_{O,a})_{ij}(\mathbf{W}_{O,b})_{kl}] = \begin{cases} \delta_{ik}\delta_{jl} & \text{for } a = b \\ c \cdot \delta_{ik}\delta_{jl} & \text{for } a \neq b \end{cases} \tag{72}$$

for correlation coefficient $c \in [-1, 1]$, where $\delta_{ij}$ is the Kronecker-delta which is 1 is $i = j$ and 0 otherwise. Under this approach, the initial magnitude of the training set logits is given by $\|\mathbf{Z}^0\|_F = \beta\sqrt{(1+c)}\|\mathbf{z}^0\|_F$, where $\|\mathbf{z}^0\|_F$ is the initial magnitude of the logits of the base model. By manipulating $\beta$ and $c$, we can independently change $\beta$ and $\|\mathbf{Z}^0\|_F$ with the caveat that $\|\mathbf{Z}^0\|_F \leq \sqrt{2}\beta\|\mathbf{z}^0\|_F$ since $c \leq 1$. It follows that the small $\beta$, large $\|\mathbf{Z}^0\|_F$ region of the phase plane (upper left in Figure 5) is inaccessible with most well-conditioned models where $\|\mathbf{z}^0\|_F \sim 1$ at initialization. If we only train one set of weights, the $\hat{\Theta}$ is independent of $c$.

Unless otherwise noted, all empirical studies in Sections 2 and 3.2 involve training a wide resnet on CIFAR10 with SGD, using GPUs, using the above correlated initialization strategy to fix $\hat{\Theta}$. All experiments used JAX (Bradbury et al., 2018) and experiments involving linearization or direct computation of the NTK used Neural Tangents (Novak et al., 2019b).

