# OpenReview forum: "Temperature check: theory and practice for training models with softmax-cross-entropy losses"
_ICLR.cc/2021/Conference — Reject_

### Official Review · AnonReviewer4 · 2020-10-18

**Rating:** 3
**Confidence:** 3

**Review:**

This paper investigates how the inverse temperature parameter $\beta$ in the softmax-cross-entropy loss impacts the learning and the generalization. In the theory part, this paper introduces the concepts of early learning timescale and nonlinear timescale, and shows how the learning dynamics depend on the parameter $\beta$. The empirical investigations are carried out with wide Resnets on CIFAR10, Resnet50 on ImageNet, and GRUs on IMDB. The results suggest that the optimal $\beta$ is architecture sensitive.

The theory developed in this paper is rudimentary. I don't see how the theory in Section 2 is crucial for the understanding of the impact of $\beta$. The results basically say proper normalization on the timescale is needed for a fair comparison among different $\beta$. There are neither formal statements nor proper discussions on the significance of the results.

From the experiment results, the performance is relatively insensitive to the parameter $\beta$ with batch normalization. I don't see a clear motivation for dropping batch normalization or tuning $\beta$ with batch normalization. It is apparent that tuning for the best $\beta$ is always no worse than setting $\beta=1$. But the impact of $\beta$ seems to be insignificant, and the so-called optimal $\beta$ varies wildly across different experiments. The performance with different $\beta$ is not presented in Sections 3.2.2 and 3.2.3.

Finally, the manuscript is poorly written and needs to be largely reworked to be considered for publication. None of the key concepts identified in the paper are formally defined. For example, it is unclear what is the mathematical meaning of $\ll$ and $\sim$ in Section 2.3, and what is the precise meaning of the phrase "no longer be neglected" in Section 2.4. Abbreviations like NTK are not introduced. Curves in the plots like Figure 6(c) dashed orange are not labeled.

---

> ### Author Response · Authors · 2020-11-14
> **Reply**
>
> First and foremost, thank you for taking the time to review our work and for your helpful comments. We believe your feedback will help us to improve our theoretical exposition. We apologize that our presentation gave you the impression that we did not have any “non-rudimentary” theoretical results. We would note that although some of the scalings seem simple (for example, the scaling of $\tau_{nl}$ with $\beta$), there is significant nuance that must be understood about the units of the problem as we discuss in sec. 2.2. We believe that a careful exposition of these nuances is valuable to the community and puts “common knowledge” on firm footing.
>
> To make things explicit, our main theoretical contributions are:
>
> * The inverse temperature $\beta$ and logit scale $||\textbf{Z}||$ control timescales which determine the rate of change of the loss, the relative change in logits, and the time for learning to leave the linear regime.
> * Generalization is generally insensitive to $||\textbf{Z}||$, except for large values which lead to poorly conditioned learning.
> * Small $\beta$ allows networks to use non-linear learning dynamics, which improve generalization.
> * The largest allowable learning rate is set by the non-linear timescale. Combined with the rate of change of the loss, this suggests that training networks with small $\beta$ requires a tradeoff between slow and unstable learning.
>
> We would like to clarify that our theory section was written informally intentionally to make the results easier to understand (for us, more formal exposition is not necessarily correlated with clarity). Thank you for your feedback, we are happy to make our exposition more formal and add explicit definitions of common notions such as `<<` and `~`. Thanks also for noting that we had not defined “NTK”, this is a great point though that is fortunately easy to correct. Indeed, all statements can be written out formally. For example, the definition of the logit change timescale $\tau_{z}$, established in Equations 6 and 7, can be more formally established as follows:
>
> $$Z(x,t) - Z(x,0) =  \textbf{v} t/\tau_{z} + O(t^2/N**1/2)$$
>
> For a vector $\textbf{v}$ with $||\textbf{v}|| = 1$ and $\tau_{z}$ as defined in Equation 7 of the paper. Other timescales can be defined similarly.
>
> Would this style of prose help you to understand the text? If so, we are happy to make the corresponding changes to the manuscript.
>
> We’d also like to directly respond to this comment:
>
> “I don't see a clear motivation for dropping batch normalization or tuning β with batch normalization.”
>
> Note that many architectures do not admit batch normalization, for example batch normalization does not improve training of RNNs or transformers. It is therefore interesting, noteworthy, and impactful that networks without batch normalization can attain similar performance to batch normalization with $\beta$ tuning.
>
> Another key point is that not all networks support batchnorm, or use batchnorm in a SOTA setting. Finally, our results on ImageNet show that combining batchnorm and $\beta$ tuning can be beneficial, even if the optimal $\beta$ is not far from $1$; the $\beta$ tuning may be a cheap way to improve performance on a difficult task.

---

> ### Author Response · Authors · 2020-11-24
> **New revision**
>
> We have uploaded a revision of the paper which seeks to clarify our contributions. We look forward to your feedback.

---

### Official Review · AnonReviewer3 · 2020-10-25
**This paper analyses theoretically and empirically how the temperature/scaling of the soft-max output layer affects the dynamics of the training of the neural network.**

**Rating:** 6
**Confidence:** 3

**Review:**

The soft-max layer of the neural network typically does not scale the outputs of the incoming layers, i.e., the scale is set to one. This paper analyses the how the dynamics of the training is effected by a non-unit scale $\beta$. The mathematics involves elementary calculus, and it is clear and easy to follow. The analysis primarily concerns the initial/early phases of the training, or what the authors refer to as short times --- note to authors: I think there should be a better phrasing than using "short times" --- where $\tau$ is small. The analysis is divided into two cases: linear dynamics where the Hessian/second-order-term is negligible , and the non-linear dynamics where it is not. The experiments are illustrating but inconclusive.

Additional comments and suggestions:
1. Sections 2.2 to 2.3 can be restructured to more bring out that both the linear and non-linear dynamics concern the early phases of the training --- unless of course I misread these sections.
2. Sections 3.1 and 3.2 does not use batch -normalisation. Although some readers can reasonably guess the reason, it is worthwhile to explicitly say why this is the case.
3. I don't think section 3.3 contributes to the central message of the paper. Hence, I think it can be move to the appendix, and some from the appendix can be moved to the main paper.
4. It is not mentioned how the first plot of Figure 2 and the two plots in Figure 3 are obtained.

The paper deserves an *accept* because it is fundamentally correct and it is one of the "secrets-of-the-trade", and I am glad that it is written. It is *not a stronger accept* because I feel the "right units for comparison" (see section 4) could already be common knowledge among those who has looked hard at soft-max.

---

> ### Author Response · Authors · 2020-11-14
> **Reply**
>
> We appreciate the comments on the structure and clarity of the text, some of which we are currently incorporating into an improved draft.
>
> We’d like to respond to this comment in particular: “ ‘The right units for comparison’ (see section 4) could already be common knowledge among those who has looked hard at soft-max.” We posit that an explicit discussion of the relationship between the different early-learning timescales is not common knowledge. Our key theoretical contributions are:
>
> * The inverse temperature $\beta$ and logit scale $||\textbf{Z}||$ control timescales which determine the rate of change of the loss, the relative change in logits, and the time for learning to leave the linear regime.
> * Generalization is generally insensitive to $||\textbf{Z}||$, except for large values which lead to poorly conditioned learning.
> * Small $\beta$ allows networks to use non-linear learning dynamics, which improve generalization.
> * The largest allowable learning rate is set by the non-linear timescale. Combined with the rate of change of the loss, this suggests that training networks with small $\beta$ requires a tradeoff between slow and unstable learning.
>
> In particular, correctly accounting for the relative timescales gives rise to subtle results. For example, the “raw” value of $\tau_{nl}$, the non-linear timescale (that is, for fixed learning rate $\eta$ and varying $\beta$) goes as $1/\beta$ - which would suggest that models with small $\beta$ are more linear. In fact, the opposite is true. The power of our analysis is in combining concepts like the effective learning rate with our explicit discussion of timescales, which we believe is not well-known.

---

> ### Author Response · Authors · 2020-11-24
> **New revision**
>
> We have uploaded a revision of the paper, incorporating some of your suggestions. We look forward to your feedback on these changes, and to our comments below.

---

### Official Review · AnonReviewer2 · 2020-10-27
**Topic is interesting, but theoretical contribution is unclear**

**Rating:** 5
**Confidence:** 2

**Review:**

This paper studies how temperature scaling affects training dynamics of neural networks (with softmax layer and cross-entropy loss). The theoretical analysis shows that neural networks trained with smaller inverse temperatures (beta) exit the linear regime faster, which implies better performance. Experiments on image classification and sentiment analysis confirm that tuning temperature improves neural network generalization, even for state-of-the-art models.

Strengths:
* The topic is novel and interesting.
* Empirical results confirm the importance of tuning temperature.

Weaknesses:
* Theoretical contribution seems unclear.
* Experiments are limited to two tasks.

I am leaning towards rejecting the paper. My biggest concern is about the theoretical contribution. The main conclusion of the theory seems to be that smaller beta is better, because it helps neural networks exit linear regime faster. But this doesn’t explain the experiment results: optimal beta varies a lot across models, and it is sometimes quite large (beta >= 1). The paper empirically shows that small beta causes more instability as an explanation, but there is no theoretical explanation. Therefore, it is unclear how to use the current theory.

Ideally, I hope the theory can be extended to explain why small beta causes instability (the conclusion section mentions this as future work), and/or how neural network architecture affects optimal beta, but these extensions do not seem obvious.

The experiments can also be expanded. Currently, there are only two tasks in the experiments. While the results are impressive, I would be more convinced if there are more tasks/models. For example, I wonder what the optimal beta is for state-of-the-art BERT-based models.  What about structured prediction tasks? Does the size of training set affects optimal beta?

On the positive side, I think the topic is novel and interesting, and the current empirical results are solid. If the theory can be extended to explain the tradeoff between small/large beta and the role of architecture, I would recommend this paper. Alternatively, the paper can also be improved by expanding experiments.

Other suggestion and question:
- Section 2 (theory) may be easier to read if there is a short summary of main claims/results.
- For IMDb experiment, is there a reason for choosing GRU instead of more recent BERT-based models?
- Some of the figure fonts are too small.

Feedbacks after author response:
I am maintaining my rating after reading the author response. It is a close decision. I like the topic, but I think the draft still has room for improvement to become a great paper. The updated draft is much clearer and answers some of my questions. Most importantly, the updated theory section explains why small beta can be bad: it slows training. While I appreciate the clarification, I think this argument still doesn't fully align with the experiment results. As the CIFAR-10 experiment points out, using small beta can still lead to good performance, so the main problem for small beta seems to be instability (rather than slow training), which the current theory couldn't explain. To improve the theory, I hope the paper can provide more insights into the instability caused by small beta. For example, I wonder if this is somehow connected to the slow training argument; perhaps the failed runs indeed suffer from slow training.

---

> ### Author Response · Authors · 2020-11-14
> **Reply**
>
> We thank the reviewer for their comments, particularly about the clarity of the conclusions in the theory section. We will take their and reviewer 4’s comments into account to improve our exposition.
>
> To summarize, our major theoretical contributions are as follows:
> * The inverse temperature $\beta$ and logit scale $||\textbf{Z}||$ control timescales which determine the rate of change of the loss, the relative change in logits, and the time for learning to leave the linear regime.
> * Generalization is generally insensitive to $||\textbf{Z}||$, except for large values which lead to poorly conditioned learning.
> * Small $\beta$ allows networks to use non-linear learning dynamics, which improve generalization.
> * The largest allowable learning rate is set by the non-linear timescale. Combined with the rate of change of the loss, this suggests that training networks with small $\beta$ requires a tradeoff between slow and unstable learning.
>
> In particular, we apologize that the final point was not made more clear. In the text we presented the empirical results on optimal learning rate $\eta^*(\beta)$ and then provided a theoretical explanation; in fact, the theory inspired us to use a $1/\beta$ scaling in the search for $\eta^*$, which turned out to be nearly exact in practice.
>
> We feel that the above results are significant; does our discussion clarify the impact of our ideas? We look forward to hearing from you, and are happy to further clarify any additional points.
>
> We agree that the architecture dependence of the optimal $\beta$ remains non-trivial, both empirically and theoretically. However, from a practical standpoint, the range of optimal $\beta$ observed across all model classes is small enough ($10^{-2}-10^{1}$) that it should be easy in practice to tune $\beta$ (see first response to reviewer 1 for an explicit tuning setup). The theory gives two actionable “rules of thumb”:
> * For large $\beta$, it is important to keep $||\beta\textbf{z}||$ small ($<10^{2}$) to avoid bad conditioning.
> * Training time and instability both increase rapidly for very small $\beta$ (e.g. corresponding to $||\beta\textbf{z}||<<10^{-2}$).
>
> Together, this suggests that tuning around the value of $\beta$ which gives $||\beta\textbf{z}|| = 1$ is effective in practice. The optimal value $\beta^*$ is hard to predict, but the theory suggests that there is not a large range which needs to be searched
> ($10^{-2}-10^{1}$ in practice).
>
> We chose GRUs on the IMDB task in order to evaluate models without pretraining, to more directly compare between a natural language task and an image classification one. We hope to extend the analysis to SOTA language models in future work.

---

> > ### Comment · AnonReviewer2 · 2020-11-20
> > **Follow-up question**
> >
> > Thank you for clarification! This is a helpful summarization.
> >
> > I'm still not 100% about what's the downside of using a small beta though (perhaps I'm missing something). My current understanding is that small beta empirically causes unstable training. Is there other theoretical reasons?
> >
> > I look forward to the updated draft!

---

> > > ### Author Response · Authors · 2020-11-20
> > > **Small $\beta$ leads to slow training**
> > >
> > > In addition to the instability, networks with smaller $\beta$ tend to train more slowly. This is predicted by our theory as follows.
> > >
> > > The maximum allowable learning rate occurs when $\tau_{nl}$ is close to $1$ (corresponding to the gradient changing significantly after a single SGD step). This gives an upper bound on the optimal learning rate of $\eta^{*} \leq c/\beta$, for an architecture-dependent, $\beta$-independent constant $c$.
> > >
> > > If we compute $\tau_{z}$, the time it takes the logits to change significantly, at the optimal learning rate, we get $\tau_{z}\approx ||\textbf{Z}^{0}||/ \beta$. What this means is that it takes  $O(1/\beta)$ SGD steps for any significant learning to take place - so networks with smaller $\beta$ will take more SGD steps to train.
> > >
> > > We found this trend to hold in our empirical work; the best results for small $\beta$ were achieved with $\eta^{*} \approx 0.01/\beta$ (Figure 4), at the cost of slow training (in number of SGD steps) for small $\beta$. We also observed empirically that training in this high learning rate regime made for less stable dynamics. The stability of small $\beta$ networks can be improved by lowering the learning rate, at the cost of both training speed (which is already significantly slower for e.g. $\beta < 10^{-3}$), but also the generalization performance of the best-performing networks.

---

> > > > ### Comment · AnonReviewer2 · 2020-11-20
> > > > **Thanks for clarification**
> > > >
> > > > Thank you. This clarifies my question.

---

> > > > > ### Author Response · Authors · 2020-11-24
> > > > > **New draft**
> > > > >
> > > > > We have uploaded a revision of the paper, and have tried to incorporate your suggestions.

---

### Official Review · AnonReviewer1 · 2020-10-29
**Introducing the novel time scaling analysis for the cross entropy loss with temperature**

**Rating:** 6
**Confidence:** 1

**Review:**

##  Summary of the paper
This paper proposed the new time scaling analysis for the cross-entropy loss with temperature in the deep neural networks. The authors introduced the effective time scale, and then provided the linear and non-linear time scale based on the output of logits with temperature and stepsizes by using the NTK theory.

## Strong and weak points of the paper
### Strong points
- Provided the novel time scaling analysis for the cross-entropy loss with many empirical validations.
### Weak points
- The choice of optimal $\beta$ still remains unclear and left to the future work.  I think clarifying the tuning strategy of $\beta$ is the most important from practical view point.

## Rating
- Clarity: The authors should clarify which part corresponds to the previous work proposed or this work proposed
- Correctness: I did not check all the proof in detail.
- Novelty: Since I am unfamiliar with this field, I thought that introducing the linear and non-linear time scale in Sec 2.3 and 2.4 seems intereting tools in experimetns.

## Comments and Questions
- Q) In Sec 2.2 to 2.4, what types of models and datasets are used in the numerical experiments ? Although I might overlooked, it should be explained explicitly.

- Q) As far as I understood, from the ovservation of Sec 2.2 to 2.4, setting large $\beta$ makes the times scales $\tau_z$ and $\tau_{nl}$ smaller, which means that  the early linear learning timescale and nonlinear timescale smaller. So, setting large $\beta$ enhance escaping from the linear dynamics at the beggining, but the time period of the nonlinear dynamics become shorter, which results in the worse final performace. Is my understanding correct ?

- Q) What is the definition of optimal $\eta^*$ in Sec 3.1?

- Following is just a comment. In Sec 3.3, the authors found that using small $\beta\in [10^{-2},1)$ might improve the final performance in a variety of networks.  On the other hand, when we consider Bayesian deep learning, using cold posterior that uses the temperatrue below $1$ results in better final perfomance compared to using $\beta=1$ (For example see, Wenzel, Florian, et al. "How good is the bayes posterior in deep neural networks really?." arXiv preprint arXiv:2002.02405 (2020). ). So, I just thought that there might be some connection between this small temperature phonomena.

---

> ### Author Response · Authors · 2020-11-14
> **Reply**
>
> First, thank you for taking the time and effort to review our work and for your detailed comments.
>
> The experiments in sections 2.2 to 2.4 are on CIFAR10, and we apologize for not making that clear. We believe you are correct about your understanding of $\beta$, but to make things more explicit, setting large $\beta$ has two main effects:
> * The network is _prevented_ from escaping the linearized regime (as seen by the increased ratio of $\tau_{nl}/\tau_{z}$).
> * If accompanied by an increase in $\beta\textbf{z}$, the scale of the logits, the learning dynamics are poorly conditioned.
> Both of these effects can degrade generalization.
>
> We appreciate the connection you have raised to results in Bayesian deep learning where $\beta>1$ is generally a good training regime. This is a great question and we will include a discussion of the relationship between our setting and Bayesian deep learning in a revision. Thanks also for pointing out Wenzel, Florian et al. which seems to provide a very interesting perspective on temperature in the Bayesian setting. A difference between their setting and ours seems to be that Bayesian deep learning relies on generating samples from a stochastic process, where the final scale of the network outputs (divided by temperature) determine the sharpness of the distribution. By contrast, in SGD we are ultimately concerned with point-estimates based on the logits, so the temperature affects the learning dynamics, but not the sampling procedure. Note that in fig. 5c we observe the final logit scale to be similar across the $\beta$-$||\textbf{Z}^{0}||$ plane indicating that the final logit scale learned by SGD is consistent across initializations.
>
> The optimal learning rate $\eta^*$ is defined as the learning rate with the best generalization performance while training wide resnet on CIFAR10 over a sweep of \beta, with $||\textbf{Z}^{0}|| = 0$. We note that the optimal learning rate analysis combined with the generalization analysis gives us our practical prescription for tuning $\beta$: at large $\beta$ (such that $\textbf{Z}^{0}||>>1$), generalization performance is generally poor. For very small $\beta$ (in practice, for values $10^{-3}$ or lower), learning is slow and unstable.
>
> The practitioner can tune a sweep over $\beta$ within the limitations of their computational budget. Suppose the budget allows for 4 additional trainings after hyperparameter tuning. Then, the model can be trained over the range $\beta\in[10^{-2},10^{-1},10^{0},10^{1}]$, using the learning rate $\eta_{0}/\beta $ (where $\eta_{0}$ is the optimal learning rate for $\beta = 1$). If there is a significant improvement at the upper or lower ends of that range, then the practitioner can perform additional $\beta$ tuning to improve the model. If there are no significant improvements, then $\beta = 1$ is likely optimal/close to optimal.
>
> We hope this clarifies the tuning procedure, and if it doesn’t we’d be happy to explain further.

---

> ### Author Response · Authors · 2020-11-24
> **New draft of paper**
>
> We have uploaded a revision of the paper. We look forward to hearing your thoughts on the changes, as well as on our response to your original comments.

---

### Decision · Program_Chairs · 2021-01-07
**Final Decision**

**Decision:**

Reject

**Comment:**

Three reviewers are mildly positive, while one is negative. The substantive comments of the reviewers are consistent with each other; it is merely their evaluations that differ.

One contribution of the paper is that it shows how using temperature tuning can yield similar accuracy to using batch normalization; this is useful because batch normalization is not always possible. The revised paper shows improvements, and we appreciate the engagement of the authors with the reviewer comments. However, there are remaining weaknesses such as a weak argument based on the empirical.results.

This paper can be improved based on the comments made by the reviewers. We encourage the authors to resubmit to a future venue.